# PARALLEL TEST-TIME SCALING WITH MULTI-SEQUENCE VERIFIERS

## ABSTRACT

Parallel test-time scaling, which generates multiple candidate solutions for a single problem, is a powerful technique for improving large language model performance. However, it is hindered by two key bottlenecks: accurately selecting the correct solution from the candidate pool, and the high inference latency from generating many full solutions. We argue that both challenges are fundamentally linked to verifier calibration. A well-calibrated verifier not only improves answer selection, but also enables early-stopping strategies to reduce latency. However, existing verifiers are limited as they score each candidate in isolation, overlooking rich contextual information across the set of candidates. To address this, we introduce the Multi-Sequence Verifier (MSV), the first verifier designed to jointly process all candidate solutions and model their interactions. MSV achieves state-of-the-art calibration, which directly enhances best-of-N selection performance. We further introduce a streaming MSV variant that empowers a novel early-stopping framework. Our novel framework fully leverages parallel decoding, which contrasts with the existing multi-sequence early exit works that decode sequences one by one and thus incur significant latency. In this novel setting, MSV can achieve the same target accuracy with around half the latency that would be required with its counterpart that scores each solution in isolation.

## 1 INTRODUCTION

Large language models (Brown et al., 2020) have become increasingly powerful, with much of their performance unlocked by test-time scaling strategies. One of the most effective strategies is parallel scaling, where a model generates multiple, independent candidate solutions for a single problem (Wang et al., 2022; Lightman et al., 2023; Snell et al., 2024). Parallel scaling can be especially fruitful (Snell et al., 2024) when complemented with sequential scaling methods (Wei et al., 2022; Jaech et al., 2024; Guo et al., 2025) that search for and revise solutions adaptively inside a chain of thought. However, parallel scaling faces two bottlenecks that limit its usage: (1) the selection problem, as accurately identifying the correct solution from a large pool of candidates is difficult, and (2) the high inference latency required to generate numerous full solutions.

We argue that these two bottlenecks are not independent; their solutions are deeply connected through the principle of calibration. The selection problem is fundamentally a classification task, where a verifier (Cobbe et al., 2021; Lightman et al., 2023) must accurately estimate the correctness of each solution. Its ability to do so, its calibration (Guo et al., 2017; Kadavath et al., 2022), directly determines the performance of downstream parallel scaling methods such as best-of-$N$. Concurrently, the high cost of generation can be mitigated by early stopping, a technique that also hinges on a well-calibrated verifier to score intermediate answers and terminate decoding once a threshold score is exceeded (Zhang et al., 2025a; Yang et al., 2025). Although this approach has only been explored in the single-sequence setting, we propose in our paper a way to extend it to parallel scaling. Thus, a verifier with superior calibration is the key to solving both the accuracy and efficiency challenges of parallel scaling.

A fundamental limitation of existing verifiers is their isolated approach to scoring, which overlooks the rich contextual information available across a full set of generated outputs. The success of self-consistency (Wang et al., 2022; Lyu et al., 2025), which relies on the simple cross-sequence statistic of vote counting, serves as a crucial insight: global statistics of sequences can be highly predictive of

individual correctness. Yet, this insight has remained surprisingly underexplored. We generalize this principle and introduce the Multi-Sequence Verifier (MSV), the first verifier model designed to learn from the interactions between all candidate solutions. By jointly processing multiple outputs, MSV produces highly accurate and calibrated scores, achieving state-of-the-art performance on standard calibration metrics.

**M**ulti-**S**equence **V**erifier (MSV)'s superior calibration directly translates into improved performance on both the parallel scaling bottlenecks. First, it enhances best-of-$N$ decoding, leading to a more accurate final answer selection and more reliable confidence scores for the chosen answers. Second, and more significantly, we introduce a streaming variant of MSV that empowers a novel framework for early stopping with parallel decoding. We decode multiple sequences in parallel while the streaming MSV calibrates their intermediate answers in real-time by jointly observing all sequences. Decoding terminates the moment any one sequence's confidence exceeds a threshold.

To be concrete, our experiments show that on challenging math reasoning benchmarks, MSV improves best-of-64 accuracy by over 6% relative to strong weighted-voting baselines. This accuracy gain is accompanied by a dramatic improvement in the calibration of the selected answer's confidence score, with MSV reducing the Expected Calibration Error by over 75%. The benefits are even more notable in our parallel early-stopping framework, where our streaming MSV achieves the same peak accuracy as baseline verifiers models with as little as half the latency. These downstream improvements stem directly from MSV's superior verifier calibration, where it reduces error metrics like the Brier score by 50% compared to models that score sequences in isolation.

Our contributions can be summarized as follows:

1. We propose the Multi-Sequence Verifier (MSV), a novel verifier architecture that models cross-sequence interactions to achieve state-of-the-art calibration.

2. We demonstrate that improved calibration from MSV directly translates to enhanced best-of-N selection and its calibration, a primary application of parallel scaling.

3. We generalize an existing early-stopping framework to the parallel decoding setting, which theoretically enables one to scale test-time compute in such a way that latency doesn't grow. We also introduce a streaming MSV variant that outperforms strong baselines in this novel setting.

## 2 RELATED WORK

### 2.1 CALIBRATION OF LLM OUTPUTS

Confidence estimation for LLMs has been studied through black-box and white-box approaches. Black-box approaches have focused on prompting a language model to verbalize its confidence (Lin et al., 2022; Tian et al., 2023). Xiong et al. (2023) finds that white-box approaches to calibration, which instead probe internal states (Kadavath et al., 2022), generally outperform their black-box counterparts. Lyu et al. (2025) investigates using global sequence statistics such as the entropy of answer distribution to calibrate the majority voted answer, but their method cannot be applied to calibrating the other answers. Zhang et al. (2025a) demonstrates an interesting result that intermediate answers from reasoning models such as DeepSeek R1 (Guo et al., 2025) can be robustly calibrated through lightweight probes on their hidden states, and apply this finding to early stopping in the decoding of a single sequence.

### 2.2 MULTI-SEQUENCE ANSWER SELECTION

Answer selection among multiple decoded sequences is a crucial component of parallel test-time scaling. One can regard this task as well-calibrated ranking of answers within a problem such that the correct answers are scored higher than the incorrect ones. As such, the most common approach is to train calibrated verifier models (Cobbe et al., 2021; Zhang et al., 2024). Self-consistency (Wang et al., 2022) is a simple selection method that doesn't require training of a verifier model, and instead selects the answer with the most numerous occurrence among sequences. This hints at the possible usefulness of global statistics, or in other words looking at multiple sequences together, in improved calibration and answer selection. Tangentially related is multi-agent debate (Du et al., 2023), since

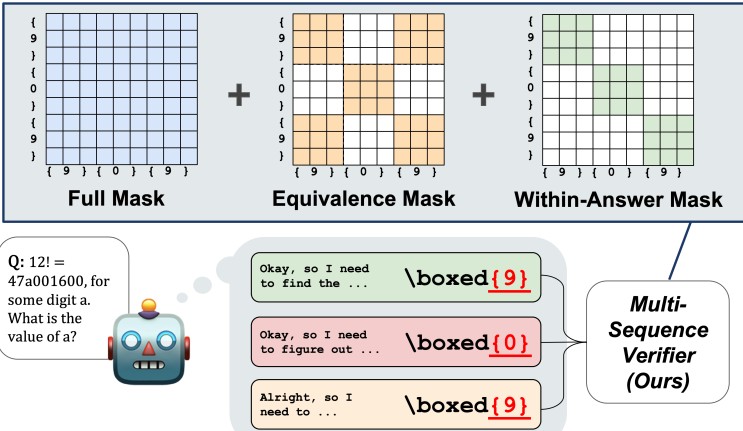

**Figure 1:** Illustration of our **M**ulti-**S**equence **V**erifier (MSV) that uses diverse attention masks in its Multi-Mask Transformer Block (MMTB) to predict the correctness of each answer. The different attention masks allow MSV to flexibly leverage both information across and within sequences.

debate with a single round is equivalent to best-of-$N$ selection with critics that look at all generated sequences to classify their own sequence as correct or incorrect. Again, the effectiveness of debate hints at the usefulness of cross-sequence features in estimating individual correctness.

### 2.3 EFFICIENT PARALLEL SCALING

The efficiency of parallel test-time scaling is essential for its practical usage. Most works on efficient test-time scaling focus on reducing the total test-time compute, but largely disregard the tradeoff between latency and throughput. It is desirable, in practical settings, to reduce latency by buying a larger number of throughput, even if the total amount of computation remains the same. Various works such as DeepConf (Fu et al., 2025) and others (Li et al., 2024; Aggarwal et al., 2023; Xue et al., 2023; Huang et al., 2025) propose early stopping at the sequence level to reduce the number of decoded sequences. However, they all assume multiple sequences being decoded sequentially, and as a result, incur tens to hundreds times the latency of a single sequence. We focus on the parallel decoding setting token-level early stopping. To the best of our knowledge, we are the first in the LLM decoding literature to investigate this setting. Our paper aims at improving over simple but strong baselines in this novel setting.

## 3 METHOD

### 3.1 PROBLEM SETUP

In this paper, we assume a *parallel decoding* or *parallel sampling* scenario, where our LLM model generates $N$ sequences simultaneously for a given query $q$. At each decoding step, one token is sampled for every sequence, in parallel. This parallel approach reduces latency compared to sequential decoding, a crucial advantage for practical applications. Under this scenario, we consider two settings: 1) **Terminal Answers** and 2) **Streaming Answers**. In brief, the **Terminal Answers** setting focuses on the final answers obtained from each sequence after decoding has been complete. The **Streaming Answers** setting focuses also on intermediate answers produced while decoding. We describe each setting in more detail below.

**Terminal Answers.** Given a query $q$, the LLM decodes $N$ parallel sequences until they all reach termination. To extract an explicit answer, we append an elicitation prompt such as `### Final Answer ### \boxed` to the end of each $n$th sequence, prompting the model to produce a boxed answer $a^{(n)}$. This results in one answer per sequence, denoted by $\{a^{(n)}\}_{n=1}^{N}$.

**Streaming Answers.** Unlike the **Terminal Answers** setting, we additionally extract *intermediate answers* whenever a delimiter, e.g., the token "Wait", is encountered. This approach of extracting intermediate answers at delimiters has been explored in prior work for single-sequence early stopping (Zhang et al., 2025a; Yang et al., 2025), which we generalize to the parallel decoding setting with multiple sequences. Specifically, when the $k$th delimiter appears in the $n$th sequence, we immediately branch that sequence, append the elicitation prompt ### Final Answer ### \boxed, and obtain an intermediate answer $a_k^{(n)}$. We also extract the terminal answer at the end of each sequence. Thus, if a sequence contains $K^{(n)} - 1$ delimiters, it may yield multiple answers $\{a_k^{(n)}\}_{k=1}^{K^{(n)}}$, $a_{K^{(n)}}^{(n)}$ is the terminal answer. For notational consistency, in the **Terminal Answers** setting, we regard the single terminal answer as $a_1^{(n)}$ and $K^{(n)} = 1$.

Along with each answer, we store the representations of the answer tokens for later use with our Multi-Sequence Verifier. Specifically, if $a_k^{(n)}$ consists of $L_k^{(n)}$ tokens, we store $\{h_{k,i}^{(n)}\}$ for $i = 1, \ldots, L_k^{(n)}$, where $h_{k,i}^{(n)} \in \mathbb{R}^d$ denotes the $d$-dimensional hidden state of the $i$th token, output by the last transformer layer of the LLM.

**Correctness and learning objectives.** Now, we define the correctness of an answer $a_k^{(n)}$ generated from a sequence. Let $a^*$ denote the ground-truth answer for the query $q$, and the symbol $\sim$ be an equivalence relation that captures symbolic or semantic equality, e.g., "$2 + 2$" $\sim$ "$4$", that can be computed with an algorithm such as SymPy checker (Meurer et al., 2017; Lewkowycz et al., 2022). Then, we define the correctness of each candidate answer $a_k^{(n)}$ as follows:

$$y_k^{(n)} = \mathbf{1}\left[a_k^{(n)} \sim a^*\right].$$

Our goal is to accurately predict $y_k^{(n)}$ for every candidate, so that we can make more effective use of the generated answers in downstream parallel test-time scaling methods such as best-of-$N$.

Departing from methods that classify each answer in isolation, we propose to *leverage information across multiple sequences* to further improve the predictions. Specifically, in the **Terminal Answers** setting, the prediction for $a_1^{(i)}$ may depend on the entire set of terminal answers $\{a_1^{(n)}\}_{n=1}^N$. In contrast, in the **Streaming Answers** setting, predictors must respect causality: the correctness prediction for $a_k^{(n)}$ may only use information from tokens generated *up to that point in time* across all sequences. A detailed description of our method is provided in the following section.

### 3.2 MULTI-SEQUENCE VERIFIER

In this section, we present our novel verifier architecture, MSV. The verifier predicts the correctness of each answer by consuming that answer tokens' (last-layer) hidden states and *aggregating* auxiliary signals drawn from the answers produced by the other sequences. This cross-sequence aggregation enables MSV to judge the correctness of a single answer in the context of diverse peer outputs, leading to better-calibrated predictions than when relying on information from a single answer alone. Refer to Fig. 1 to see the schematic diagram of MSV. We also provide a pseudocode for MSV's forward pass in Algorithm 1.

**Input representation for MSV.** Let $t$ be a time index shared across parallel sequences, where all the sequences start generating from $t = 0$ and they generate one token at a time. Let $\tau_k^{(n)}$ denote the time stamp at which the $n$th sequence generates the $k$th answer $a_k^{(n)}$. At a specific readout time step $t$, MSV takes the representations of all the answers generated up to the step $t$ to predict the correctness of the answers. Specifically, we collect the representations as follows,

$$U^{(t)} = \text{concat}\left(\left\{\left\{\left\{h_{k,i}^{(n)} + \mathsf{e}^{(n)}\right\}_{i=1}^{L_k^{(n)}}\right\}_{\tau_k^{(n)} \leq t}\right\}_{n=1}^N\right).$$

Here, $\text{concat}(\cdot)$ denotes concatenation of all the representations along the sequence dimension, without any special separators. $\mathsf{e}^{(n)}$ is a learnable per-sequence embedding added to identify token representations from different sequences.

**Multi-mask transformer blocks.** To process the aggregated representation $U^{(t)}$, we introduce a **M**ulti-**M**ask **T**ransformer **B**lock (MMTB). This block captures diverse aspects of the token sequences by combining multiple attention outputs, each derived from a different mask applied to the

---

**Algorithm 1** Multi-Sequence Verifier (MSV) Forward Pass

---

**Require:** $N$ sequences with hidden states $\{h_{k,i}^{(n)}\}$, readout time $t$

**Ensure:** Correctness predictions $\{\tilde{y}_k^{(n)}\}$ for all answers up to time $t$

**Trainable parameters**: Sequence embeddings $\{e^{(n)}\}$, QKV projections $(W_Q, W_K, W_V)$, mask weights $\{\mathbf{w}_h\}$, MLP & LN, feature MLP, prediction head $(\mathbf{w}, b)$

1: $U^{(t)} \leftarrow \text{concat}\left( \left\{ \left\{ \{h_{k,i}^{(n)} + e^{(n)}\}_{i=1}^{L_k^{(n)}} \right\}_{\tau_k^{(n)} \leq t} \right\}_{n=1}^{N} \right)$

2: Compute query, key, value projections: $(Q_h^{(t)}, K_h^{(t)}, V_h^{(t)})_{h=1}^{H}$

3: **for** each head $h = 1, \dots, H$ and mask $j = 1, \dots, J$ **do**

4:      $A_{h,j}^{(t)} \leftarrow \text{softmax}\left( \frac{Q_h^{(t)}(K_h^{(t)})^\top}{\sqrt{d}} + \log M_j \right) V_h^{(t)}$

5: **end for**

6: $\tilde{U}^{(t)} \leftarrow \sum_{h=1}^{H} \sum_{j=1}^{J} \alpha_{h,j} A_{h,j}^{(t)}$     where $(\alpha_{h,1}, \dots, \alpha_{h,J}) = \text{softmax}(\mathbf{w}_h)$

7: $Z^{(t)} \leftarrow (U^{(t)} + \tilde{U}^{(t)}) + \text{MLP}(\text{LN}(U^{(t)} + \tilde{U}^{(t)}))$

8: **for** each answer $a_k^{(n)}$ generated by time $t$ **do**

9:      Compute $\gamma_k^{(n)} \leftarrow \frac{1}{N} \sum_{m=1}^{N} \mathbf{1}[a_k^{(m)} \sim a_k^{(n)}]$

10:     $\bar{z}_k^{(n)} \leftarrow z_{k,L_k^{(n)}}^{(n)} + \text{MLP}(\gamma_k^{(n)})$

11: **end for**

12: **if** Terminal Answers mode **then**

13:     $\tilde{y}_1^{(n)} \leftarrow \sigma\left( \frac{1}{|\mathcal{C}(n)|} \sum_{m \in \mathcal{C}(n)} (\mathbf{w}^\top \bar{z}_1^{(m)} + b) \right)$ for all $n$

14: **else if** Streaming Answers mode **then**

15:     $\tilde{y}_k^{(n)} \leftarrow \sigma(\mathbf{w}^\top \bar{z}_k^{(n)} + b)$ for all $(n, k)$

16: **end if**

17: **return** $\{\tilde{y}_k^{(n)}\}$

---

same input, $U^{(t)}$. As in standard multi-head attention (Vaswani et al., 2017), we begin by computing the usual linear projections of $U^{(t)}$ into query, key, and value matrices $(Q_h^{(t)}, K_h^{(t)}, V_h^{(t)})_{h=1}^{H}$ where $H$ is the number of heads. For a fixed collection of $J$ masks $\{M_j\}_{j=1}^{J}$, we then compute the output for each mask:

$$A_{h,j}^{(t)} = \text{softmax}\left( \frac{Q_h^{(t)}(K_h^{(t)})^\top}{\sqrt{d}} + \log M_j \right) V_h^{(t)},$$

where $\log M_j$ adds 0 to permitted entries and $-\infty$ to masked entries, with $\text{softmax}$ applied row-wise. The masked outputs are then combined via learnable mixture weights $\mathbf{w}_h = (w_{h,1}, \dots, w_{h,J})$ for each head $h$:

$$\tilde{U}^{(t)} = \sum_{h=1}^{H} \sum_{j=1}^{J} \alpha_{h,j} A_{h,j}^{(t)}, \qquad (\alpha_{h,1}, \dots, \alpha_{h,J}) = \text{softmax}(\mathbf{w}_h).$$

The number of masks, $J$, is a fixed hyperparameter for both training and inference. In our instantiation, we use four complementary masks. The *full* mask permits all interactions, $(M_{\text{full}})_{u,v} = 1$ for all positions $u, v$, enabling attention across all tokens in all sequences. The *within-sequence* mask restricts attention to tokens originating from the same sequence,

$$(M_{\text{ws}})_{u,v} = \mathbf{1}\big[\text{seq}(u) = \text{seq}(v)\big],$$

capturing within-sequence signals while blocking cross-sequence signals. Here, $\text{seq}(u)$ denotes the index $n$ of the sequence containing the token $u$. The *equivalence* mask allows attention only between tokens whose answers are symbolically equivalent,

$$(M_{\text{eq}})_{u,v} = \mathbf{1}\big[\text{ans}(u) \sim \text{ans}(v)\big],$$

where $\text{ans}(u)$ is the answer $a_k^{(n)}$ containing the token $u$.

Finally, the *within-answer* mask allows attention only between tokens inside a single answer instance $a_k^{(n)}$,

$$(M_{\text{wa}})_{u,v} = \mathbf{1}\big[(\text{seq}(u), \text{step}(u)) = (\text{seq}(v), \text{step}(v))\big],$$

where $\text{step}(u)$ identifies the answer step $k$ to which the token belongs.

To summarize, $a_k^{(n)}$ can attend to $a_{k'}^{(n')}$ (1) through the full mask, (2) through the within-sequence mask if $n = n'$, (3) through the equivalence mask if $a_k^{(n)} \sim a_{k'}^{(n')}$, and (4) through the within-answer mask if $n = n'$ and $k = k'$. In the **Terminal Answers** setting, the within-sequence mask and within-answer mask are equivalent, reducing the number of masks to three. We further restrict the attention masks to be "causal" in the **Streaming Answers** setting, meaning that an answer $a_k^{(n)}$ may attend to $a_{k'}^{(n')}$ only if $\tau_k^{(n)} \geq \tau_{k'}^{(n')}$.

The block's final output is then computed using standard residual connections and an MLP layer:

$$Z^{(t)} = (U^{(t)} + \tilde{U}^{(t)}) + \text{MLP}\big(\text{LN}(U^{(t)} + \tilde{U}^{(t)})\big),$$

Here, MLP denotes a Multi-Layer Perceptron and LN is layer normalization, consistent with the standard self-attention plus MLP architecture.

**Feature augmentation.** While the aggregated representation $Z^{(t)}$ has undergone the MMTB module which updates token representations by modeling relationships across sequences and among symbolically equivalent answers, we further enrich each answer's representation with an explicit statistic: the proportion of sequences that produce symbolically equivalent answers. Although attention over all tokens could, in principle, internalize such counting signals, exact counting is known to be challenging for Transformer architectures (Barbero et al., 2024; Ouellette et al., 2023). Motivated by this, we compute the fraction $\gamma_k^{(n)} \in [0, 1]$ of sequences whose answers are equivalent to $a_k^{(n)}$, and inject it as a learned feature by projecting it through a small MLP which is then added to the hidden states. Specifically, from $Z^{(t)}$, we extract the representation $z_{k,L_k}^{(n)}$ corresponding to the last token of $a_k^{(n)}$, and add the information from $\gamma_k^{(n)}$, as follows:

$$\bar{z}_k^{(n)} = z_{k,L_k^{(n)}}^{(n)} + \text{MLP}(\gamma_k^{(n)}).$$

We only use the the last token hidden state for computing the prediction, because the attention layer in MMTB has already aggregated the relevant information from all the other tokens in $a_k^{(n)}$.

**Constructing final predictions.** To predict the final correctness of $a_k^{(n)}$, we apply a linear head to its augmented representation and pass the resulting logit through a sigmoid:

$$\tilde{y}_k^{(n)} = \sigma\big(\mathbf{w}^\top \bar{z}_k^{(n)} + b\big),$$

where $\sigma$ denotes the sigmoid function and $\mathbf{w}, b$ are learnable parameters. Additionally, in the **Terminal Answers** setting, instead of computing each $\tilde{y}_1^{(n)}$ independently, we first average the *logits* of symbolically equivalent terminal answers and then apply the sigmoid function as follows,

$$\tilde{y}_1^{(n)} = \sigma\left(\frac{1}{|\mathcal{C}(n)|} \sum_{m \in \mathcal{C}(n)} \big(\mathbf{w}^\top \bar{z}_1^{(m)} + b\big)\right) \text{ where } \mathcal{C}(n) := \{m \in \{1, \ldots, N\} \mid a_1^{(m)} \sim a_1^{(n)}\}.$$

Logit averaging is inappropriate in the case of **Streaming Answers**, because the information carried by $a_k^{(n)}$ grows with $k$, and logits at earlier steps might disrupt rather than complement the prediction at the current step $k$.

**Training and Inference.** For the **Streaming Answers** scenario, we run the sequences until all of them terminate, collect all the intermediate and final answers produced, and train the parameters to minimize the binary cross-entropy loss. Let $T$ be the global time step when all the sequences terminate. We first compute $Z^{(T)}$, compute the predictions, and minimize

$$\mathcal{L} := \sum_{n,k} \text{BCE}(\tilde{y}_k^{(n)}, y_k^{(n)}),$$

where the summation is over all valid $n, k$ pairs that represent answers. For the **Terminal Answers** scenario, we proceed similarly, except that we minimize BCE only for the terminal answers. BCE is a strictly proper loss, which encourages calibrated probabilities under standard assumptions. (Blasiok et al., 2023)

At inference time, whenever we want to predict whether the sequences are producing correct answers at specific time $t$, we compute $Z^{(t)}$ as described above, and predict $\{\{\tilde{y}_k^{(n)}\}_{\tau_k^{(n)} \leq t}\}_{n=1}^N$.

### 3.3 APPLICATIONS

In this section, we present how to use the MSV-based correctness predictions $\tilde{y}_k^{(n)}$ in the **Terminal Answers** and **Streaming Answers** settings. Briefly, in the **Terminal Answers** setting, we use $\tilde{y}_1^{(n)}$ as a verifier score for *best-of-N* decoding, whereas in the **Streaming Answers** setting, we use $\tilde{y}_k^{(n)}$ both as an early-stopping signal and as a criterion for selecting the best candidate. Detailed descriptions are provided below.

**Calibrated best-of-$N$ predictions.** In the **Terminal Answers** scenario, our verifier scores are used for *best-of-N* decoding, a robust technique for improving performance (Nakano et al., 2021). From $N$ candidate answers, we select the one, $a^\dagger$, with the highest score:

$$n^\dagger = \arg\max_n s_1^{(n)}, \qquad a^\dagger = a_1^{(n^\dagger)}.$$

Our Multi-Sequence Verifier (MSV) naturally provides these scores by predicting the correctness probability $p_1^{(n)}$ for each answer. We therefore simply set the score $s_1^{(n)} := p_1^{(n)}$.

A key advantage of this approach is that the score for the chosen answer, $\tilde{y}^\dagger = p_1^{(n^\dagger)}$, is not just a ranking tool but also a calibrated confidence estimate (Guo et al., 2017). This allows us to assess not only *which* answer is best but also *how likely* it is to be correct.

**Early stopping of parallel decoding.** In the **Streaming Answers** setting, multiple sequences are decoded in parallel, and each intermediate answer $a_k^{(n)}$ yields a correctness score $\tilde{y}_k^{(n)}$. We define an early-stopping rule that halts decoding once the best available candidate exceeds a threshold $\lambda$:

$$t^* = \min\left\{ t : \max_{n,k: \tau_k^{(n)} \leq t} p_k^{(n)} \geq \lambda \right\}.$$

At $t^*$, we output the candidate with the highest predicted score. If the system terminates without any of the decoded answers exceeding the threshold, the system outputs the highest score among the final answers $\tilde{y}_{K^{(n)}}^{(n)}$:

$$n^\dagger = \arg\max_n \tilde{y}_{K^{(n)}}^{(n)}, \qquad a^\dagger = a_{K^{(n)}}^{(n^\dagger)}.$$

Thus, $\tilde{y}_k^{(n)}$ acts both as the stopping signal and as verifier scores for selecting the best answer.

## 4 EXPERIMENTS

### 4.1 EXPERIMENTAL SETUP

In this section, we present empirical evidence that demonstrates the effectiveness of MSV, our novel verifier that leverages information across all sequences. We denote MSV trained on groups of $N$ sequences, by $\text{MSV}_N$. For strong baselines that classify each sequence in isolation, or *single-sequence baselines* for short, we consider two methods: 1) "$\text{MSV}_1$", trained only on each sequence, and 2) "Probe" (Zhang et al., 2025a), which trains an MLP on last-token representations to predict the correctness of each answer. Although we can directly use these two methods as verifiers in the multiple-sequence setting, both are single-sequence predictors and hence can be complemented by aggregation methods that combine verifier scores across sequences. To this end, we adopt a strong aggregation method called Weighted Voting (WV; Li et al., 2022), which aggregates correctness probabilities by summing them within each class of equivalent answers and normalizing across classes, extending the single-sequence baselines for fair comparison with $\text{MSV}_N$. In the

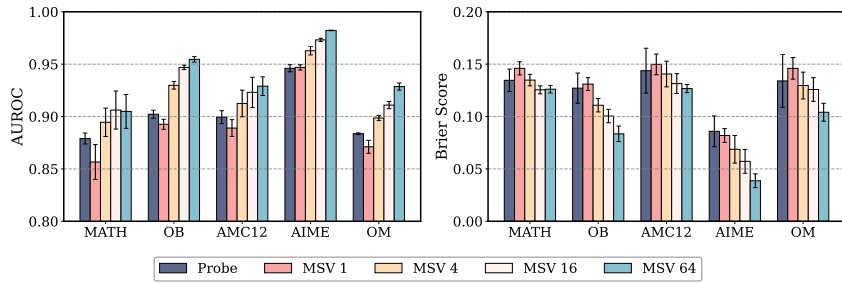

**Figure 2:** Comparison of baselines and $MSV_N$ on AUROC and Brier Score in the **Terminal Answers** setting. OB refers to OlympiadBench, and OM refers to Omni-MATH.

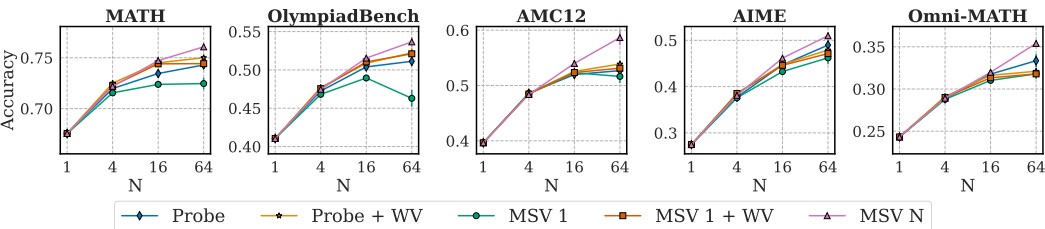

**Figure 3:** Accuracy of best-of-$N$ with Probe and $MSV_1$, both with and without WV, and $MSV_N$.

experiments, this extension is denoted as $A$+WV, where $A$ represents either Probe or $MSV_1$. We assess calibration with AUROC, Expected Calibration Error (ECE; Naeini et al., 2015), and Brier Score (BS; Brier, 1950), where higher AUROC and lower ECE/BS indicate better performance. Details are provided in § A.

We use DeepSeek-R1-Distill-Qwen-1.5B model (Guo et al., 2025) as the base LLM. DeepMath-103K (He et al., 2025) is used for training and validation, and evaluation is conducted on MATH (Hendrycks et al., 2021), OlympiadBench (OB; He et al., 2024), AMC12, AIME, and Omni-MATH (OM; Gao et al., 2024) datasets. All experiments are done on five random seed values. We provide further details of the experimental setup in § A, along with a full report in § C. In § B, we report additional experiments with other base LLM, with open-source verifiers, with training-free methods such as self-consistency, and with other delimiters, along with ablations on masks in MMTB and detailed wall-clock time measurements.

### 4.2 TERMINAL ANSWERS SETTING

In this section, we conduct experiments in the **Terminal Answers** setting. We first empirically demonstrate that $MSV_N$ predicts the correctness of each candidate answer more accurately than baseline methods, as evidenced by their improvements on the standard calibration metrics. We then show that, in the parallel scaling scenario, using $MSV_N$ as a verifier ensures better accuracy and calibration compared to the baselines.

**Calibration on Terminal Answers.** Calibration of verifiers is presumably the key determinant of effective parallel scaling. To verify the calibration ability, we first evaluate on the fundamental task of scoring individual candidate answers. Specifically, the metrics reported here are computed over *all* generated sequences in the test set. Fig. 2 reports the AUROC and BS of the single-sequence baselines and $MSV_N$ on the five evaluation datasets. We find that $MSV_N$ consistently achieves better calibration compared to all baselines, and that the performance scales reliably as $N$ increases. On the AIME dataset, $MSV_{64}$ achieves around 50% reduced BS compared to Probe. Our results clearly demonstrate that leveraging information across multiple sequences can indeed be used to improve the calibration of verifiers.

**Calibrated Best-of-$N$ Prediction.** We apply $MSV_N$, our superior verifier model that leverages cross-sequence attention, to best-of-$N$ (BoN) decoding and calibrated confidence outputs. As shown in Fig. 3, $MSV_N$ consistently improves BoN accuracy over all baselines, with gains becoming more

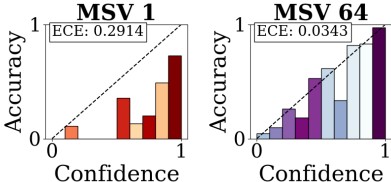

**Figure 4:** Probability estimates for a single problem with 16 completions, comparing $MSV_1$, weighted voting applied to $MSV_1$, and $MSV_{16}$.

**Figure 5:** Reliability diagrams of verifier confidence of on best-of-64 answers (AIME).

**Table 1:** ECE of confidence output by verifiers on their *best-of-64 answers*. Lower ECE implies better calibration.

| Method | MATH | AIME | OM |
|---|---|---|---|
| Probe | $0.185_{\pm0.028}$ | $0.301_{\pm0.059}$ | $0.419_{\pm0.080}$ |
| $MSV_1$ | $0.213_{\pm0.014}$ | $0.316_{\pm0.035}$ | $0.450_{\pm0.030}$ |
| $MSV_{64}$ | $\mathbf{0.132}_{\pm0.015}$ | $\mathbf{0.075}_{\pm0.050}$ | $\mathbf{0.130}_{\pm0.027}$ |

**Table 2:** Brier score of verifiers trained and evaluated on Streaming Answers. Lower Brier score implies better calibration.

| Method | MATH | AIME | OM |
|---|---|---|---|
| Probe | $0.159_{\pm0.010}$ | $0.080_{\pm0.011}$ | $0.153_{\pm0.024}$ |
| $MSV_1$ | $0.160_{\pm0.013}$ | $0.064_{\pm0.003}$ | $0.138_{\pm0.008}$ |
| $MSV_{64}$ | $\mathbf{0.115}_{\pm0.002}$ | $\mathbf{0.032}_{\pm0.001}$ | $\mathbf{0.126}_{\pm0.005}$ |

pronounced as $N$ increases. At $N = 64$, $MSV_{64}$ achieves a $\sim6\%$ improvement over the strongest single-sequence baselines, while alternative methods often plateau or even deteriorate. Given that WV methods perform the closest to $MSV_N$, one might wonder if the behavior of $MSV_N$ is almost identical to WV methods. Fig. 4 shows the predictions of $MSV_1$, $MSV_1$ + WV, and $MSV_{16}$ on an instance with one correct sequence. Contrary to that hypothesis, $MSV_{16}$ behaves in a different manner from WV, which incorrectly suppresses the correct answer due to its low voting count of one.

Beyond accuracy, calibration of the verifier's confidence on the BoN answer is crucial, especially in risk-sensitive applications where decisions rely not only on the predicted answer but also on its associated confidence. Among calibration metrics, ECE is particularly important, as it measures how well predicted confidence aligns with the true frequency of correctness. Table 1 shows that $MSV_{64}$ consistently lowers ECE across all datasets, most notably reducing it to just 25% of Probe on AIME. Reliability diagrams in Fig. 5 further confirm that $MSV_{64}$ outputs more diverse and well-aligned confidence estimates, and additional metrics in § B indicate similar improvements.

In summary, modeling cross-sequence interactions with $MSV_N$ not only boosts BoN accuracy but also delivers substantially more reliable and calibrated confidence estimates, reinforcing its utility for downstream decision-making.

### 4.3 STREAMING ANSWERS

In this section, we conduct experiments in the **Streaming Answers** setting. Unlike **Terminal Answers**, this setting involves early stopping, so accurately predicting the correctness for intermediate answers is directly tied to both the performance and the efficiency. We first examine how much $MSV_N$ improves calibration over the baselines in the **Streaming Answers** setting, and then empirically validate how this improved calibration contributes to the effectiveness of early stopping.

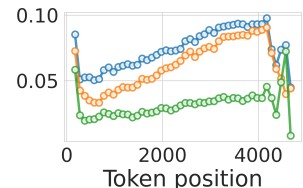

**Figure 6:** Brier score of verifiers per token position bin, on the AIME dataset. Blue, Orange, and Green represent Probe, $MSV_1$, and $MSV_{64}$, respectively.

**Calibration on Streaming Answers.** Table 2 provides the Brier scores of baselines and $MSV_{64}$ in the streaming answers settings. From the results, we see that, similar to the case of terminal answers, $MSV_N$ significantly improves over baselines in calibration. Notably, the Brier score on AIME is more than halved by $MSV_{64}$ compared to the single-sequence baselines. Fig. 6 plots the BS calculated per bins that contain answers in certain ranges of token positions. The figure shows that $MSV_{64}$ improves over baselines at

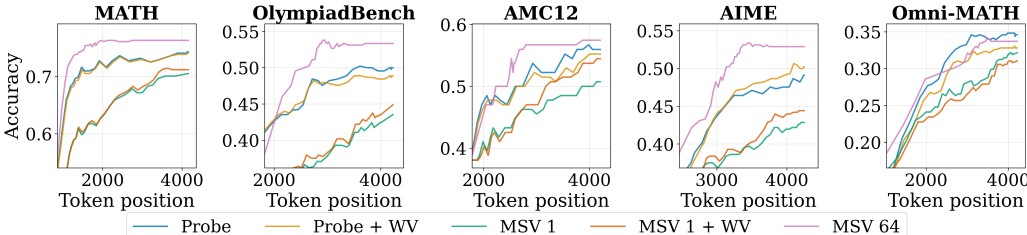

**Figure 7:** Accuracy-latency tradeoff curves of parallel early stopping with $N = 64$ parallel sequences. We plot the tradeoff curves for $MSV_1$, $MSV_1$ plus weighted voting, and $MSV_{64}$. Curves that lie higher above are superior in that they achieve higher accuracy given the same latency budget, and also in that they require lower latency to achieve the same target accuracy.

every range of token positions, without exception. Furthermore, the gap in performance tends to increase for answers that appear later in the sequences.

**Parallel Early Stopping with Streaming Answers.** Next, we investigate whether the superior performance of $MSV_N$ in classifying the correctness of streaming answers can indeed transfer to improved performance in parallel early stopping as proposed in § 3.3. We compare $MSV_N$ with $MSV_1$, and also the WV method adopted to the streaming answers setting, explained in detail in § A.1. Recall that early stopping is done with some threshold value $0 \leq \lambda \leq 1$. At each threshold $\lambda$, we can plot the accuracy of the early stopped answers and the latency, which we approximate with the early stopped token position. Therefore, sliding $\lambda$ between 0 and 1 creates an accuracy-latency tradeoff curve for each verifier.

In Fig. 7, we plot the accuracy-latency tradeoff curves of $MSV_1$, $MSV_1$ with WV, and $MSV_{64}$ when performing early stopping across datasets with $N = 64$ parallel sequences. First, we observe that weighted voting doesn't help as much as in best-of-$N$, which is presumably due to the fact that the answers given at early timestamps that are aggregated at later timestamps are of bad quality, often deviating significantly from the ground truth answer. Most importantly, we observe significant improvement of $MSV_{64}$ over the single sequence baselines. On all the datasets, we find that the maximum achievable accuracy with $MSV_1$ can be achieved by $MSV_{64}$ with a smaller latency, the reduction ranging from 50% to 75%. Both the best-of-$N$ results and the parallel early stopping results support our hypothesis that improved calibration of verifiers effectively transfer to their downstream performance in parallel scaling methods. We have only use the token position as an approximation of latency, and the actual latency measurements can be found in § B.1, which shows that the overheads incurred by verifier inference and intermediate answer generation are minimal.

## 5 CONCLUSION

In this work, we addressed two critical bottlenecks in the parallel test-time scaling of large language models: the accurate selection of correct solutions and the high inference latency from generating multiple candidates. We argued that both challenges are fundamentally linked to verifier calibration and that existing verifiers are limited by scoring candidate solutions in isolation. To overcome this, we introduced the Multi-Sequence Verifier (MSV), a novel architecture designed to jointly process an entire set of candidate solutions and model their interactions. Our extensive experiments demonstrated that MSV achieves state-of-the-art calibration, significantly outperforming strong baselines that score sequences independently. This improved calibration directly translated to substantial downstream benefits. For best-of-N decoding, MSV improved selection accuracy by over 6% and reduced the Expected Calibration Error of the final chosen answers by over 75%.

Furthermore, we introduced a novel parallel early-stopping framework, a new setting for efficient inference that contrasts with prior sequential approaches. In this framework, a streaming variant of our MSV achieved the same peak accuracy as baseline verifiers with around half the latency. These findings underscore the importance of cross-sequence information and establish a new, more effective approach to building and utilizing verifiers for parallel scaling.

REPRODUCIBILITY STATEMENT.

We present detailed description of the MSV architecture in § 3.2. Further details regarding the algorithm's hyperparameters, training, baselines, datasets, models, and metrics used in experiments are provided in § A. We also provide a reproducible codebase in the supplementary materials.

ETHICS STATEMENT.

We propose a new method that improves calibration of verifiers for improved downstream parallel scaling performance. Although our approach doesn't have a direct positive or negative impact in ethical or societal aspects, it improves and accelerates task completions with LLMs through parallel scaling. This could be used for good, such as reliable decision-making in problems that benefit the society. However, like many technologies, this method could also be misused in domains where improved decision-making can be used with bad intentions. Therefore, researchers and practitioners should apply these methods with consideration of their broader societal implications, aiming to ensure that the benefits of the technology are used responsibly and ethically.

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

# A  EXPERIMENTAL DETAILS

## A.1  DETAILS ON BASELINE METHODS

### A.1.1  SINGLE-SEQUENCE BASELINES

To show that the interaction between sequences does really benefit the classification performance of MSV, we compare MSV trained on $N > 1$ against MSV trained on $N = 1$, which serves as a controlled baseline. We abbreviate MSV trained on $N$ sequences simply as $\text{MSV}_N$. As another baseline, we follow Zhang et al. (2025a) and train a 2-layer MLP "Probe" on the last token representations $h_{k,L_k^{(n)}}^{(n)}$ to predict the correctness of answers $a_k^{(n)}$. We find that the Probe serves as a strong baseline that often matches or even exceeds the performance of $\text{MSV}_1$.

We focus on white-box methods in our paper, and do not consider verifiers in the generative family (Zhang et al., 2024). This is because their training and inference costs lie on a different order of magnitude compared the white-box methods that only require one forward pass. Their training scheme is also very different. Altogether these factors make such methods inappropriate to compare against MSV which operates on the hidden states of LLMs with a single forward pass per prediction.

There are also training-free methods for calibration, such as using the base LLM's token probabilities on the answer tokens (Yang et al., 2025). Specifically, we use the geometric mean of probabilities over all tokens in an answer, to estimate the answer's correctness.

$$s_k^{(n)} = \left( \prod_{i=1}^{L_k^{(n)}} p_{\text{LLM}} \left( a_{k,i}^{(n)} \mid \text{tokens}_{<a_{k,i}^{(n)}} \right) \right)^{1/L_k^{(n)}}.$$

### A.1.2  AGGREGATION BASELINES

We can further complement the single-sequence baselines with aggregation methods that combine predictions across multiple sequences through simple heuristics. Although prior work has not investigated explicit aggregation of single-sequence baselines for improving the classification of individual candidates, we introduce a baseline, *Weighted Voting* (WV), inspired by the weighted best-of-$N$ approach (Li et al., 2022), as follows: for each equivalence class of symbolically identical answers, we aggregate its answers' correctness probabilities by summing them, and normalize the probabilities across equivalence classes. Performing best-of-$N$ with the WV probabilities is equivalent to weighted best-of-$N$, which picks the equivalence class with the biggest aggregate probability. In the streaming setting, we can also perform weighted voting for candidate $a_k^{(n)}$ by aggregating and normalizing over the symbolically equivalent one that come before time $\tau_k^{(n)}$. However, when done naively, this can disrupt the predictions at the early token positions, due to a lack of other existing candidates. For instance, weighted voting will always assign a probability of one to the first output candidate, since there are not other candidates to normalize over. Therefore, we set a threshold $R \in \mathbb{Z}$ such that weighted voting is performed only when the number of candidates exceeds $R$ at time $\tau_k^{(n)}$. We find that $R = 16$ yields good overall performance and use this setting throughout our experiments with WV in the **Streaming Answers** setting.

We also propose a training-free baseline inspired by *Self-Consistency* (Wei et al., 2022) that only uses the vote count (number of symbolically identical answers) to score each answer. Specifically, we score each answer $a_k^{(n)}$ with

$$s^{(n)} = \sum_{m=1}^{N} \mathbf{1}[a_1^{(n)} \sim a_1^{(m)}]/N$$

for Terminal Answers. For the **Streaming Answers** setting, we similarly use the fraction of symbolically identical answers among all that have been decoded until time $\tau_k^{(n)}$, again using the threshold $R = 16$ to prevent miscalibration at the early candidates. Note that the self-consistency baseline is equivalent to the weighted voting baseline with a verifier that outputs a constant score.

## A.2 DETAILS ON DATASETS AND MODELS

The pre-trained model and datasets used in our experiments are summarized below.

**Models.**

- `DeepSeek-R1-Distill-Qwen-1.5B`: https://huggingface.co/deepseek-ai/DeepSeek-R1-Distill-Qwen-1.5B, licensed under MIT License.[1]

**Datasets.**

- `MATH-500`: https://huggingface.co/datasets/HuggingFaceH4/MATH-500, licensed under MIT License.[2]
- `Olympiad Bench`: https://huggingface.co/datasets/Hothan/OlympiadBench, licensed under MIT License.[3]
- `AMC12`: https://huggingface.co/datasets/rulins/amc12_22-24, not openly licensed; usage is subject to MAA AMC policies.[4]
- `AIME`: https://www.kaggle.com/datasets/hemishveeraboina/aime-problem-set-1983-2024, licensed under CC0 License. [5]
- `Omni-MATH`: https://huggingface.co/datasets/KbsdJames/Omni-MATH, dataset files licensed under Apache-2.0 License.[6]
- `DeepMath-103K`: https://huggingface.co/datasets/zwhe99/DeepMath-103K, licensed under MIT License.[7]

We use the DeepSeek-R1-Distill-Qwen-1.5B model (Guo et al., 2025) as our base LLM that generates the reasoning traces and answers to reasoning problems. All training data for our verifier models are generated with the base LLM on problems in the DeepMath-103K dataset, which underwent careful decontamination of public reasoning evaluation datasets (He et al., 2025). We evaluate our model on the MATH dataset (Hendrycks et al., 2021), the AMC12 problems between 2022 and 2024, the AIME problems between 1983 and 2024, OlympiadBench (He et al., 2024), and Omni-MATH (Gao et al., 2024). We use 224 and 64 randomly chosen problems from the DeepMath-103K as our training and validation sets, respectively. An evaluation dataset with size larger than 448 is randomly subsampled to 448 problems. We generate 64 responses from the base LLM on every problem. This creates training and validation sets with 14K and 4K sequences, respectively, and evaluation datasets with at most 28K sequences.

## A.3 DETAILS ON METRICS

We evaluate verifier predictions on four standard probabilistic metrics. Let $\{(p_i, y_i)\}_{i=1}^{m}$ denote the dataset of $m$ predicted probabilities and their corresponding binary correctness labels.

**Area Under the Receiver Operating Characteristic Curve (AUROC).** The Area Under the Receiver Operating Characteristic Curve (AUROC) measures the probability that a randomly chosen positive example receives a higher score than a randomly chosen negative example:

$$\text{AUROC} = \Pr\left[p^+ > p^-\right] + \frac{1}{2}\Pr\left[p^+ = p^-\right],$$

where $p^+$ and $p^-$ are the predicted probabilities for a random positive and negative sequence, respectively.

---

[1]https://huggingface.co/deepseek-ai/DeepSeek-R1-Distill-Qwen-1.5B/blob/main/LICENSE
[2]https://github.com/openai/prm800k/blob/main/LICENSE
[3]https://github.com/OpenBMB/OlympiadBench/blob/main/LICENSE
[4]https://maa.org/student-programs/amc/maa-american-mathematics-competitions-policies/
[5]https://www.kaggle.com/datasets/hemishveeraboina/aime-problem-set-1983-2024
[6]https://huggingface.co/datasets/KbsdJames/Omni-MATH
[7]https://github.com/zwhe99/DeepMath/blob/main/LICENSE

**Brier score.** The Brier score (Brier, 1950) is the mean squared error between predicted probabilities and binary outcomes:

$$\text{Brier} = \frac{1}{m} \sum_{i=1}^{m} (p_i - y_i)^2.$$

Lower values indicate better calibrated and sharper probabilities.

**Negative Log-Likelihood (NLL).** The negative log-likelihood, or log loss, evaluates the quality of probabilistic predictions:

$$\text{NLL} = -\frac{1}{m} \sum_{i=1}^{m} \Big( y_i \log p_i + (1 - y_i) \log(1 - p_i) \Big).$$

**Expected Calibration Error (ECE).** The Expected Calibration Error (Naeini et al., 2015; Guo et al., 2017) measures the discrepancy between a model's predicted confidences and the actual accuracies. To compute ECE, predictions are grouped into $J$ bins based on their confidence scores. The ECE is the weighted average of the absolute difference between the mean confidence and the mean accuracy within each bin:

$$\text{ECE} = \sum_{j=1}^{J} \frac{|B_j|}{m} \left| \text{acc}(B_j) - \text{conf}(B_j) \right|,$$

where $B_j$ is the set of indices of predictions whose confidence $p_i$ falls into the $j$-th bin, $\text{acc}(B_j) = \frac{1}{|B_j|} \sum_{i \in B_j} y_i$ is the accuracy of that bin, and $\text{conf}(B_j) = \frac{1}{|B_j|} \sum_{i \in B_j} p_i$ is the average confidence of that bin. We use $J = 10$ in all our reports of ECE.

**Best-of-$N$ Accuracy.** Best-of-$N$ (BoN) Accuracy is the primary metric for evaluating the effectiveness of the verifier-guided selection process in the Terminal Answers setting. For each problem, the system generates $N$ candidate answers. A verifier then provides a score for each candidate, and the one with the highest score $p^\dagger$, denoted $a^\dagger$, is selected as the final output. The accuracy is the fraction of problems in an evaluation set $Q$ for which this selected answer is equivalent to the ground truth answer $a^*$. Formally, it is expressed as:

$$\text{Accuracy} = \frac{1}{|Q|} \sum_{q \in Q} y_q^\dagger, \quad \text{where } y_q^\dagger = \mathbf{1}[a_q^\dagger \sim a_q^*]$$

and $a_q^\dagger$ is the verifier-selected answer for problem $q$, and $a_q^*$ is the ground truth. A higher accuracy indicates a more effective verifier that is better at identifying the correct solution from the pool of candidates, thus serving as a direct measure of downstream task performance.

**Best-of-$N$ ECE and Brier score.** Best-of-$N$ ECE and Brier scores are the ECE and Brier scores computed over the set $\{(p_q^\dagger, y_q^\dagger)\}$ best-of-$N$ confidence and correctness pairs.

**Area Under the Tradeoff Curve.** Sliding the threshold $\lambda$ between 0 and 1 shifts the point on the accuracy-latency tradeoff curve. One can readily show that latency increases monotonically with increasing threshold $\lambda$, due to more conservative early stopping. Moreover, the latency at $\lambda = 0$ and $\lambda = 1$ are constant with fixed dataset and $N$. It is thus appropriate to quantify the aptness of a verifier at early stopping with a single scalar value, namely the area under the tradeoff curve (AUTC). Importantly, a verifier that achieves higher accuracy than another verifier at every latency will also be characterized by a higher value of AUTC. This is also logically equivalent to requiring lower latency to achieve the same target accuracy.

A.4 DETAILS ON EXPERIMENTS

We adopt the setup and terminology of Terminal/Streaming answers from § 3, and follow the experimental protocol of § 4.1. Below we provide additional details necessary for reproducibility.

| Category | Setting (ours) |
|---|---|
| Optimizer | AdamW |
| LR schedule | `constant-with-warmup` (HF defaults) |
| LR sweep | $\{1e{-}5, 5e{-}5, 1e{-}4, 5e{-}4, 1e{-}3, 5e{-}3\}$ (CV on validation) |
| Final LRs | Probe: $1e{-}3$; Others: $5e{-}5$; $w$: $1e{-}1$; $e^{(n)}$: $1e{-}3$ |
| Probe arch. | 2-layer MLP, hidden size 1024 |
| MSV depth | 1 Multi-Mask Transformer Block |
| Batch | 64 |
| Gradient accumulation steps | 1 |
| Epochs | Terminal: 1; Streaming: 2 (4 only during LR sweep) |
| Gradient clipping | `max_grad_norm` $= 1.0$ |
| Gradient checkpointing | Not used |
| Max sequence length | 4096 |
| Temperature | 1.0 |
| Answer length | $\leq 40$ tokens (after elicitation) |

**Table 3:** Summary of implementation and hyperparameters.

**Table 4:** Token counts required to achieve the target accuracy on MATH.

| $N$ | **Probe** | **MSV$_N$** |
|---|---|---|
| 4 | 3699.5 | 2886.3 |
| 16 | 4022.8 | 1616.4 |
| 64 | 4131.8 | 1407.4 |

***Optimizer and schedule.*** All models are trained with AdamW, combined with the Hugging Face default `constant-with-warmup` scheduler (default warmup ratio). Gradient clipping is applied with `max_grad_norm` set to 1.0.

***Learning rate selection.*** We sweep learning rates in $\{1e{-}5, 5e{-}5, 1e{-}4, 5e{-}4, 1e{-}3, 5e{-}3\}$ and choose the best value by cross-validation on the validation split.

***Final learning rates.*** The probe MLP is trained with lr $= 1e{-}3$. All other modules use lr $= 5e{-}5$, except the MMTB mixture weights $w$ (lr $= 1e{-}1$) and the per-sequence embeddings $e^{(n)}$ (lr $= 1e{-}3$).

***Model capacity.*** The MSV verifier consists of a single Multi-Mask Transformer Block (one transformer layer) whose width and attention heads follow the base model. The probe is implemented as a 2-layer MLP with hidden size 1024.

***Batching and epochs.*** We use a global batch size of 64 with no gradient accumulation. Training runs for 1 epoch on Terminal data and 2 epochs on Streaming data. For LR selection, up to 4 epochs are used on the validation split.

***Decoding and prompting.*** We follow the elicitation protocol in the main paper, using the boxed `Final Answer` format and the "`Wait`" delimiter for Streaming. Decoding uses temperature $= 1.0$ without further heuristics. Each output is allowed up to 4096 tokens, with the elicited final answer truncated to 40 tokens.

***Hardware.*** All experiments fit within a single RTX A6000 GPU (48 GB memory) per model.

## B ADDITIONAL EXPERIMENTS

### B.1 LATENCY ANALYSIS IN THE STREAMING ANSWERS SETTING

To substantiate our claim that MSV improves the accuracy-latency tradeoff compared to baselines, we provide detailed measurements of latency on the MATH dataset in the Streaming Answers setting. For each verifier and $N$, we measure the minimum latency at which the respective accuracy matches or surpasses the highest accuracy achievable by Probe (the target accuracy). We break down

**Table 5:** Average latency of CoT generation on MATH (seconds).

| $N$ | **Probe** | **MSV$_N$** |
|---|---|---|
| 4 | 87.5 | 68.3 |
| 16 | 175.4 | 70.4 |
| 64 | 568.7 | 193.7 |

**Table 6:** Average latency of intermediate answer generation on MATH (seconds).

| $N$ | **Probe** | **MSV$_N$** |
|---|---|---|
| 4 | 0.9 | 0.7 |
| 16 | 1.9 | 0.7 |
| 64 | 6.2 | 2.1 |

the latency into three components: chain-of-thought generation, intermediate answer generation, and verifier inference.

**Token Efficiency.**  Table 4 reports the token positions at which each verifier reaches the target accuracy for different values of $N$. MSV$_N$ consistently achieves the same accuracy as Probe using substantially fewer tokens across all values of $N$, with the gap widening as $N$ increases.

**Wall-Clock Time Analysis.**  We measure the actual wall-clock time for each of the three components using batched computation on a single A6000 GPU. Tables 5, 6, and 7 report the average latency for chain-of-thought generation, intermediate answer generation, and verifier inference, respectively. While MSV incurs additional overhead for verifier inference due to processing multiple sequences, this overhead remains small in absolute terms (under 4 seconds even for $N = 64$) and is more than compensated by the substantial reductions in generation time. Note that there is also latency incurred by running the symbolic equivalence checker SymPy, which was on average 0.34 seconds, but omit from our calculation.

**End-to-End Latency.**  Table 8 reports the total end-to-end latency by summing the three components. MSV$_N$ achieves lower latency than Probe across all values of $N$, with reductions ranging from 21.8% to 65.4%. These results confirm that MSV not only improves token efficiency but also delivers practical wall-clock time improvements in the Streaming Answers setting.

**Training latency.**  In the Terminal Answers setting, training takes around 50 minutes regardless of the verifier. In the Streaming Answers setting, training of Probe, MSV 1, MSV 4, MSV 16, and MSV 64 takes around 105, 105, 105, 110, and 145 minutes, respectively. We performed cross-validation on the number of epochs, meaning that running more epochs on Probe, MSV 1, or MSV 4 doesn't improve their performance.

## B.2    DISCUSSION ON PARALLEL EARLY STOPPING

Although usually not the case in practice, one can increase the value of $N$ without increasing the latency of a single decoding step, e.g. by scaling the number of GPUs proportionally with $N$. In this hypothetical setting, one can "buy" more $N$ to both improve task performance *and* reduce latency compared to single-sequence decoding! Fig. 8 shows that latency of best-of-$N$ generally increases with the number of sequences $N$, because the maximum sequence length strictly increases

**Table 7:** Average latency of verifier inference on MATH (seconds).

| $N$ | Probe | $\text{MSV}_N$ |
|---|---|---|
| 4 | 0.0003 | 0.1 |
| 16 | 0.001 | 0.4 |
| 64 | 0.005 | 3.1 |

**Table 8:** Average end-to-end latency on MATH (seconds).

| $N$ | Probe | $\text{MSV}_N$ | Reduction |
|---|---|---|---|
| 4 | 88.5 | **69.2** | 21.8% |
| 16 | 177.3 | **71.6** | 59.6% |
| 64 | 575.0 | **199.0** | 65.4% |

in expectation with increasing $N$:

$$\mathbb{E}[L^{(1)}] < \mathbb{E}[\max(L^{(1)}, \ldots, L^{(4)})] < \mathbb{E}[\max(L^{(1)}, \ldots, L^{(16)})] < \ldots$$

where $L^{(i)}$ is the length of the fully decoded $i$th sequence. On the other hand, the figure shows that parallel early stopping with $\text{MSV}_N$ allows for reduction of latency *and* improved task performance, with increasing $N$. We would like to note that this claim only holds true in the hypothetical setting where latency per token position is constant regardless of $N$, and where the overhead incurred by the forward pass of $\text{MSV}_N$ is minimal. We don't claim that any of these two are true in our experimental setup. Instead, we point at the potential of our novel framework, that contrasts starkly with previous multi-sequence early stopping frameworks which introduce latency that grows about linearly with $N$—and thus wouldn't have even fit in our plots.

### B.3 ABLATION STUDY OF ATTENTION MASKS

To assess the contribution of each attention mask in MSV, we conduct an ablation study by removing each mask individually and evaluating the resulting model's calibration performance. Table 9 reports the Brier score, AUROC, and negative log-likelihood (NLL) of the Streaming $\text{MSV}_{16}$ model on AIME, with each mask ablated in turn.

The results show that all masks contribute to the overall performance, but the equivalence and within-sequence masks are especially important. Removing the equivalence mask increases the Brier score by 18.4% and substantially degrades AUROC and NLL, indicating that cross-sequence attention to equivalent answers is critical for effective verification. Removing the within-sequence mask also leads to noticeable degradation across all metrics. In contrast, removing the full mask has the smallest effect, suggesting that the more specialized masks capture the most relevant information for verification.

**Motivation for Multiple Masks.** Our decision to use multiple specialized masks was informed by preliminary experiments in which we allowed the verifier to attend to all tokens in the sequence, including chain-of-thought tokens. We initially hypothesized that, since attending to more tokens strictly increases the available information, specialized masks would be unnecessary.

Contrary to this expectation, we observed that restricting attention to only the answer tokens led to substantially better generalization. This indicated that transformers can be distracted by less critical information, and that explicitly guiding attention toward the most informative parts of the sequence improves performance. Building on this insight, we tested more specialized masks that attend only to semantically related subsets of answers (e.g., equivalent answers within and across sequences). The ablation results confirm that appropriately combining these specialized masks yields the strongest overall performance, as each mask captures complementary aspects of the multi-sequence structure.

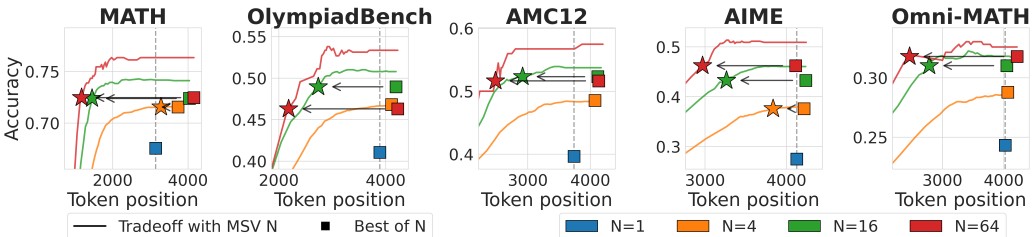

**Figure 8:** Accuracy-latency tradeoff curves of parallel early stopping with $MSV_N$, with multiple values of $N$ overlapped in a single plot. Square markers represent best-of-$N$ with the verifier $MSV_1$ trained only on terminal answers. Star markers are assigned to points on tradeoff curves that achieve the same accuracy as best-of-$N$.

**Table 9:** Mask ablation results on AIME with $MSV_{16}$ in the Streaming Answers setting.

| Metric | $MSV_{16}$ | w/o full | w/o equiv. | w/o within-seq | w/o within-ans |
|--------|------------|----------|------------|----------------|----------------|
| Brier ↓ | **0.038** | 0.040 | 0.045 | 0.048 | 0.050 |
| AUROC ↑ | **0.953** | **0.953** | 0.922 | 0.945 | 0.949 |
| NLL ↓ | **0.150** | 0.157 | 0.236 | 0.192 | 0.169 |

### B.4    ABLATION: LOGIT AVERAGING IN STREAMING ANSWERS

In § 3.2, we presented several design choices of MSV without extensive justification. The most notable is the absence of logit averaging in the prediction stage of the Streaming Answers setting. We therefore design a version of MSV that uses logit averaging during training and inference. Since the predictions have to respect causality in the Streaming Answers setting, we average the logits over all symbolically equivalent answers that come before in time. We train the the ablated model architecture with $N = 16$, and evaluate on the AIME dataset. We report the standard calibrations metrics in Table 10. One can see that logit averaging actually hurts the performance of $MSV_{16}$, potentially due to the low quality logits in the early parts of the sequences that become aggregated even in later parts of the sequences through averaging.

### B.5    MSV WITH DIFFERENT BASE LANGUAGE MODELS

In addition to the main experiments, we evaluate the robustness of MSV across different base language models and datasets. Concretely, we consider three additional base LMs: DeepSeek-R1-Distill-Llama-8B, Qwen3-1.7B in thinking mode, and Llama-3.2-1B-Instruct. Unless otherwise stated, we reuse the same training setup and hyperparameters as in the main experiments.

**DeepSeek-R1-Distill-Llama-8B on AIME.**    We first consider a larger base model, DeepSeek-R1-Distill-Llama-8B, in the AIME Terminal Answers setup. Table 11 reports calibration metrics for different verifiers, and Table 12 reports best-of-64 accuracy. Increasing $N$ consistently improves the calibration of MSV, and $MSV_{64}$ yields the highest AUROC and lowest Brier score. When used for best-of-64 selection, $MSV_{64}$ also outperforms all baselines, including Probe with and without weighted voting.

**Qwen3-1.7B (thinking mode) on AIME.**    We next evaluate Qwen3-1.7B in thinking mode on AIME. As shown in Table 13, MSV achieves progressively better calibration as $N$ increases, with substantial gains in both Brier score and AUROC compared to Probe. Table 14 shows that $MSV_{64}$ also improves best-of-64 accuracy over all baselines.

**Table 10:** Ablation of MSV architectural choices in the Streaming Answers setting. We report calibration metrics on the AIME dataset.

| Method | AUROC ↑ | Brier ↓ | NLL ↓ |
|---|---|---|---|
| $MSV_1$ | $0.906_{\pm 0.008}$ | $0.064_{\pm 0.003}$ | $0.241_{\pm 0.011}$ |
| $MSV_{16}$ | $\mathbf{0.953}_{\pm 0.007}$ | $\mathbf{0.038}_{\pm 0.003}$ | $\mathbf{0.150}_{\pm 0.011}$ |
| Ablation | $0.936_{\pm 0.0016}$ | $0.047_{\pm 0.009}$ | $0.190_{\pm 0.0333}$ |

**Table 11:** Brier score and AUROC of verifiers with DeepSeek-R1-Distill-Llama-8B on AIME.

| | Probe | $MSV_1$ | $MSV_{16}$ | $MSV_{64}$ |
|---|---|---|---|---|
| Brier ↓ | 0.0668 | 0.0770 | 0.0298 | 0.0283 |
| AUROC ↑ | 0.9668 | 0.9746 | 0.9908 | 0.9923 |

**Llama-3.2-1B-Instruct on MATH.** Finally, we consider Llama-3.2-1B-Instruct, a non-reasoning model whose accuracy on AIME is close to zero. For this model we therefore train and evaluate on the MATH train/test splits. Table 15 reports calibration metrics, and Table 16 reports best-of-64 accuracy. Even in this lower-accuracy regime, $MSV_{16}$ and $MSV_{64}$ significantly improve calibration over Probe, and $MSV_{64}$ achieves the best best-of-64 accuracy.

Overall, these results indicate that the benefits of MSV are not restricted to a particular base model or dataset: across all three additional base LMs, increasing $N$ systematically improves both calibration and best-of-$N$ accuracy.

### B.6 COMPUTATIONAL COMPLEXITY OF MSV

In the Streaming Answers setting, $N$ sequences are decoded in parallel. Assuming delimiters are uniformly distributed throughout a sequence, the total number of delimiters across all sequences—and thus the total number of forward passes through MSV—is $O(NL)$, where $L$ is the average sequence length.

The computational cost of each MSV forward pass is linear in the number of answer tokens it attends to, which is also $O(NL)$. Crucially, the cost is not quadratic because we only need to query the attention mechanism with the last token of the most recent answer. This yields a total MSV computational cost of $O(N^2L^2)$, whereas the decoding of the $N$ sequences itself scales as $O(NL^2)$. Thus, MSV adds an extra factor in $N$, while the dependence on sequence length $L$ matches that of the underlying decoding process. We address this limitation in § B.7.

### B.7 SCALING WITH MSV M FOR LARGE N

To address scalability concerns when $N$ is large (e.g., in the order of 100s), we employ a simple but effective remedy: use $MSV_M$ with $N$ parallel sequences, where $M$ is kept constant as we scale $N$. This is achieved by partitioning the $N$ sequences into $N/M$ groups, each of size $M$, and processing each group through $MSV_M$ independently. This reduces the computational complexity from $O(N^2L^2)$ to $O(MNL^2)$, or equivalently $O(NL^2)$ when treating $M$ as a constant.

Table 17 reports the area under the tradeoff curve (AUTC) in the $N = 128$ setting on the AIME dataset using this partitioning scheme. We observe that $MSV_M$ provides a better tradeoff than Probe's vanilla verification, even when $M \neq N$. Importantly, because $M$ is fixed and small, MSV's additional overhead remains at most proportional to the cost of decoding the sequences themselves, while still benefiting from joint reasoning within each group of size $M$. This makes it practical to scale $N$ beyond 64–128 without incurring a superlinear blow-up in cost relative to standard parallel decoding with vanilla verification.

**Table 12:** Best-of-64 accuracy with DeepSeek-R1-Distill-Llama-8B on AIME.

|  | **Probe** | **Probe + WV** | **$MSV_1$** | **$MSV_1$ + WV** | **$MSV_{64}$** |
|---|---|---|---|---|---|
| Best-of-64 | 0.6104 | 0.6337 | 0.6052 | 0.6386 | **0.6625** |

**Table 13:** Brier score and AUROC of verifiers with Qwen3-1.7B (thinking mode) on AIME.

|  | **Probe** | **$MSV_1$** | **$MSV_4$** | **$MSV_{16}$** | **$MSV_{64}$** |
|---|---|---|---|---|---|
| Brier $\downarrow$ | 0.0849 | 0.0770 | 0.0595 | 0.0440 | 0.0317 |
| AUROC $\uparrow$ | 0.9530 | 0.9570 | 0.9696 | 0.9825 | 0.9878 |

## B.8    SENSITIVITY TO DELIMITER CHOICE IN STREAMING ANSWERS

In the Streaming Answers setting, we use delimiter tokens to segment sequences into intermediate answers. While our main experiments use "Wait" as the delimiter, we investigate the sensitivity of MSV to this choice by evaluating two alternative delimiters: "Alternatively" and "But", both of which frequently appear as the first token of a paragraph in mathematical reasoning. On the AIME dataset, "Wait", "Alternatively," and "But" occur roughly once every 200, 900 and 400 tokens, respectively

Table 18 reports the area under the tradeoff curve (AUTC) for each delimiter-verifier combination. The results show that while the choice of delimiter does affect the absolute performance of the tradeoff, the relative ordering between verifiers remains consistent across all delimiters: $MSV_{16}$ consistently outperforms both Probe and $MSV_1$, demonstrating the robustness of our approach to delimiter selection.

## B.9    COMPARISON WITH EXTERNAL VERIFIERS.

To contextualize MSV's performance relative to commonly used verification methods in the literature, we compare against Qwen2.5-Math-PRM-7B (Zhang et al., 2025b), one of the most performant open-source verifiers. (Liu et al., 2025) We evaluate in the best-of-64 setting, where we rank all 64 terminal answers using either the external verifier or $MSV_{64}$. Additionally, we report results for Qwen2.5-Math-PRM-7B combined with weighted voting (WV).

Table 19 shows that $MSV_{64}$ consistently outperforms the external PRM across all benchmarks, with particularly substantial gains on harder datasets such as OlympiadBench, AIME, and Omni-MATH. These results demonstrate that MSV's joint reasoning over multiple candidate solutions provides advantages beyond what can be achieved with standalone process reward models, even when those models are trained on large-scale verification datasets.

## B.10    TRAINING SET SIZE

Our training set consists of 224 problems sampled from the MATH training set. While this may appear small in absolute terms, we sample 64 model responses per problem, yielding $224 \times 64 = 14,336$ sequences in total. This is substantially larger than the 1,000 sequences used by Zhang et al. (2025a) in training their probes for single-sequence early stopping.

To validate this choice, we conducted preliminary experiments with Probe in the Terminal Answers setting, varying the number of training problems. Table 20 shows that performance plateaus beyond 224 problems, with doubling to 448 problems yielding no improvement (and even a slight degradation). This saturation behavior suggests that our verifiers efficiently learn the verification task with relatively few problems, which represents a practical advantage: MSV can be trained quickly without requiring extensive problem collections or prolonged training procedures.

**Table 14:** Best-of-64 accuracy with Qwen3-1.7B (thinking mode) on AIME.

|  | **Probe** | **Probe + WV** | **MSV$_1$** | **MSV$_1$ + WV** | **MSV$_{64}$** |
|---|---|---|---|---|---|
| Best-of-64 | 0.5223 | 0.5089 | 0.4910 | 0.5294 | **0.5491** |

**Table 15:** Brier score and AUROC of verifiers with Llama-3.2-1B-Instruct on MATH.

|  | **Probe** | **MSV$_1$** | **MSV$_{16}$** | **MSV$_{64}$** |
|---|---|---|---|---|
| Brier ↓ | 0.1236 | 0.1192 | 0.0596 | 0.0510 |
| AUROC ↑ | 0.7856 | 0.7654 | 0.9222 | 0.9513 |

### B.11 LATENCY SCALING WITH PARALLEL SEQUENCES

In § 1, we claim that MSV *theoretically* enables parallel early stopping where latency does not grow with the number of sequences $N$. This claim holds in the idealized setting where computational resources (specifically, the number of GPUs) scale proportionally with $N$, such that the per-token decoding latency remains constant.

Table 21 reports the end-to-end latency in this idealized setting when using Probe on the MATH dataset. For each value of $N$, we set the threshold $\lambda$ such that the accuracy matches or surpasses the pass@1 baseline. The results show that the latency required to achieve this fixed target accuracy decreases with increasing $N$, validating our claim. However, we observe that latency saturates around $N = 64$, which is consistent with the saturation of best-of-$N$ performance at large $N$ (Snell et al., 2024).

### B.12 MSV WITH MEAN ACROSS ANSWER TOKENS

In our main experiments, MSV uses only transformer block output at the last token position of each answer in order to obtain the logit. This design choice was inherited from prior work on probes (Zhang et al., 2025a), which operate exclusively on the final hidden state. We hypothesized that the attention layer would be sufficiently expressive to aggregate relevant information from all answer tokens.

However, an alternative design is to use the mean of hidden states across all answer tokens instead of only the last token. To evaluate this variant, we compare MSV$_N$ against MSV$_N$ (Mean) in the Terminal Answers setting on the AIME dataset.

Table 22 shows that the mean variant provides a modest but consistent improvement in Brier score across all values of $N$. However, this improvement comes at a computational cost: the mean variant requires querying the attention mechanism with all answer tokens rather than just the last token, resulting in slightly higher computational overhead. Given this tradeoff and the already strong performance of the last-token variant, we use the last-token approach in our main experiments for computational efficiency.

## C FULL REPORT OF MAIN EXPERIMENTS

This section presents the full results on our main experiments, including all benchmarks, evaluation metrics, and baselines including the training-free methods. All tables and figures are provided here in complete form for reference.

**Full report on the Terminal Answers setting with trained verifiers.** Table 23 and Table 24 report all the standard calibration metrics–AUROC, BS, ECE, and NLL–in the **Terminal Answers** setting. We find that the metrics improve across all values of $N$ across all datasets, following the trend reported in § 4.2. Similarly, Table 25 reports the full best-of-$N$ accuracy across all values of $N$. We find that the best single-sequence baselines perform almost the same but slightly better than MSV$_4$ at $N = 4$, and that MSV$_N$ greatly improves the accuracy over the single-sequence baselines at $N = 16$ and 64. This is also consistent with our reports in § 4.2. Finally, we show all the

**Table 16:** Best-of-64 accuracy with Llama-3.2-1B-Instruct on MATH.

|  | **Probe** | **Probe + WV** | **MSV$_1$** | **MSV$_1$ + WV** | **MSV$_{64}$** |
|---|---|---|---|---|---|
| Best-of-64 | 0.2393 | 0.3402 | 0.1732 | 0.3402 | **0.3778** |

**Table 17:** Area Under the Tradeoff Curve on the AIME dataset with $N = 128$ parallel sequences

| **Method** | **Probe** | **MSV$_1$** | **MSV$_4$** | **MSV$_{16}$** | **MSV$_{64}$** |
|---|---|---|---|---|---|
| **AUTC** | 984.3 | 963.8 | 1060.0 | 1130.6 | 1168.9 |

calibration metrics of best-of-$N$ confidence, namely ECE and BS, in Table 26. In terms of best-of-$N$ calibration, MSV$_N$ outperform best single-sequence baselines across all values of $N$, implying much more reliable confidence levels.

**Full report on the Streaming Answers setting with trained verifiers.**  Next, we analyze the full results for the **Streaming Answers** setting. Table 27 and Table 28 show the standard calibration metrics, while Table 29 reports the Area Under the Tradeoff Curve (AUTC) for parallel early stopping. Consistent with the findings in § 4.3, the results demonstrate that $MSV_N$ consistently and significantly outperforms all single-sequence baselines across every metric and dataset. The performance improvement scales with $N$, although $MSV_{64}$ slightly deteriorates compared to $MSV_{16}$. The superior per-answer calibration of MSV$_N$ directly translates to more effective early stopping, as shown by the higher AUTC scores in Table 29. The AUTC for $MSV_N$ is substantially higher than for the baselines, quantitatively confirming that our method achieves a better accuracy-latency tradeoff.

**Full report on the training-free baselines**  Finally, we report the performance of the **training-free baselines** for completeness (Table 30 through Table 36). While methods like Self-Consistency and those based on token probabilities provide useful reference points, they are substantially outperformed by the trained verifiers across all settings. In the Terminal Answers setting, the trained baselines like Probe and $MSV_1$ are far better calibrated (Table 30 and Table 31 vs. Table 23 and Table 24). This performance gap is even more pronounced in the downstream Best-of-N application, where $MSV_N$ achieves much higher accuracy (Table 32 vs. Table 25) and produces significantly more reliable confidence scores for its chosen answers (Table 33 vs. Table 26). The training-free methods are particularly ill-suited for the Streaming Answers setting, where their poor calibration leads to very low AUTC scores (Table 36 vs. Table 29). This comprehensive comparison underscores the necessity of training a dedicated verifier to effectively leverage parallel scaling.

## USE OF LLMS

We used large language models (LLMs) as part of our experiments, but otherwise solely for polishing our writing. We didn't use them for other purposes, such as retrieving related work or generating new ideas.

**Table 18:** Area Under the Tradeoff Curve with different delimiters on the AIME dataset with $N = 16$ parallel sequences

| Delimiter | Probe | $MSV_1$ | $MSV_{16}$ |
|---|---|---|---|
| Wait | 1024.2 | 954.8 | **1078.3** |
| Alternatively | 985.9 | 980.0 | **1012.4** |
| But | 1039.3 | 1023.6 | **1070.7** |

**Table 19:** Best-of-64 performance comparison between Qwen2.5-Math-PRM-7B and $MSV_{64}$ across multiple mathematical reasoning benchmarks

| Dataset | PRM-7B | PRM-7B + WV | $MSV_{64}$ |
|---|---|---|---|
| MATH | 0.7411 | 0.7433 | **0.7607** |
| OlympiadBench | 0.3906 | 0.4018 | **0.5366** |
| AMC12 | 0.5224 | 0.5000 | **0.5866** |
| AIME | 0.4397 | 0.4174 | **0.5098** |
| Omni-MATH | 0.2344 | 0.2366 | **0.3540** |

**Table 20:** Brier score of Probe on the AIME dataset as a function of training set size

| # of problems | Brier score |
|---|---|
| 56 | 0.0833 |
| 112 | 0.0821 |
| 224 | 0.0809 |
| 448 | 0.0812 |

**Table 21:** End-to-end latency (in seconds) of early stopping required to achieve a fixed target accuracy (matching or surpassing pass@1) on the MATH dataset with Probe as the verifier, assuming the number of GPUs scales proportionally with $N$

| $N$ | Latency |
|---|---|
| 1 | 51.8 |
| 4 | 25.8 |
| 16 | 22.9 |
| 64 | 17.6 |
| 128 | 19.6 |

**Table 22:** Brier score comparison between $MSV_N$ using the last token versus the mean of answer tokens on the AIME dataset in the Terminal Answers setting.

| Method | $N = 1$ | $N = 4$ | $N = 16$ | $N = 64$ |
|---|---|---|---|---|
| $MSV_N$ | 0.0819 | 0.0687 | 0.0572 | 0.0387 |
| $MSV_N$ (Mean) | 0.0756 | 0.0641 | 0.0544 | 0.0382 |

**Table 23:** AUROC and Brier score of verifiers trained and evaluated on terminal answers.

| | MATH | | OlympiadBench | | AMC12 | | AIME | | Omni-MATH | |
|---|---|---|---|---|---|---|---|---|---|---|
| | AUROC ↑ | Brier ↓ | AUROC ↑ | Brier ↓ | AUROC ↑ | Brier ↓ | AUROC ↑ | Brier ↓ | AUROC ↑ | Brier ↓ |
| Probe | 0.8790±0.0052 | 0.1346±0.0107 | 0.9022±0.0038 | 0.1271±0.0144 | 0.8994±0.0062 | 0.1438±0.0214 | 0.9461±0.0034 | 0.0859±0.0147 | 0.8836±0.0008 | 0.1340±0.0252 |
| MSV 1 | 0.8566±0.0166 | 0.1460±0.0063 | 0.8926±0.0048 | 0.1310±0.0061 | 0.8890±0.0080 | 0.1498±0.0099 | 0.9469±0.0025 | 0.0819±0.0006 | 0.8711±0.0062 | 0.1460±0.0103 |
| Probe + WV 4 | 0.7930±0.0033 | 0.1649±0.0008 | 0.8494±0.0018 | 0.2164±0.0017 | 0.8222±0.0040 | 0.2367±0.0010 | 0.8624±0.0037 | 0.2100±0.0017 | 0.8251±0.0025 | 0.2704±0.0028 |
| MSV 1 + WV 4 | 0.7920±0.0023 | 0.1656±0.0005 | 0.8479±0.0009 | 0.2168±0.0013 | 0.8204±0.0026 | 0.2371±0.0014 | 0.8636±0.0012 | 0.2096±0.0008 | 0.8219±0.0012 | 0.2711±0.0021 |
| MSV 4 | **0.8945**±0.0135 | **0.1348**±0.0055 | **0.9298**±0.0036 | **0.1108**±0.0064 | **0.9125**±0.0127 | **0.1406**±0.0123 | **0.9628**±0.0040 | **0.0687**±0.0132 | **0.8986**±0.0025 | **0.1296**±0.0128 |
| Probe + WV 16 | 0.8228±0.0018 | 0.1416±0.0007 | 0.9028±0.0022 | 0.1588±0.0010 | 0.8728±0.0030 | 0.1759±0.0025 | 0.9092±0.0055 | 0.1368±0.0039 | 0.8714±0.0026 | 0.1921±0.0006 |
| MSV 1 + WV 16 | 0.8228±0.0033 | 0.1425±0.0003 | 0.9008±0.0010 | 0.1612±0.0008 | 0.8691±0.0029 | 0.1786±0.0018 | 0.9057±0.0024 | 0.1404±0.0015 | 0.8672±0.0010 | 0.1946±0.0008 |
| MSV 16 | **0.9062**±0.0182 | **0.1255**±0.0038 | **0.9469**±0.0021 | **0.1005**±0.0064 | **0.9231**±0.0144 | **0.1315**±0.0095 | **0.9732**±0.0015 | **0.0572**±0.0114 | **0.9109**±0.0034 | **0.1258**±0.0113 |
| Probe + WV 64 | 0.8346±0.0030 | 0.1370±0.0009 | 0.9173±0.0026 | 0.1466±0.0014 | 0.8844±0.0050 | 0.1649±0.0033 | 0.9216±0.0066 | 0.1217±0.0050 | 0.8830±0.0029 | 0.1764±0.0012 |
| MSV 1 + WV 64 | 0.8364±0.0019 | 0.1380±0.0005 | 0.9130±0.0014 | 0.1499±0.0009 | 0.8829±0.0034 | 0.1682±0.0023 | 0.9168±0.0031 | 0.1264±0.0018 | 0.8788±0.0012 | 0.1794±0.0008 |
| MSV 64 | **0.9049**±0.0162 | **0.1261**±0.0036 | **0.9546**±0.0026 | **0.0835**±0.0074 | **0.9290**±0.0089 | **0.1267**±0.0039 | **0.9822**±0.0003 | **0.0387**±0.0066 | **0.9286**±0.0034 | **0.1041**±0.0085 |

**Table 24:** ECE and NLL of verifiers trained and evaluated on terminal answers.

| | MATH | | OlympiadBench | | AMC12 | | AIME | | Omni-MATH | |
|---|---|---|---|---|---|---|---|---|---|---|
| | ECE ↓ | NLL ↓ | ECE ↓ | NLL ↓ | ECE ↓ | NLL ↓ | ECE ↓ | NLL ↓ | ECE ↓ | NLL ↓ |
| Probe | 0.0726±0.0401 | 0.4391±0.0485 | 0.0700±0.0517 | 0.4172±0.0437 | 0.1064±0.0581 | 0.4680±0.0730 | 0.1024±0.0483 | 0.3115±0.0488 | 0.1206±0.0589 | 0.4394±0.0817 |
| MSV 1 | 0.0849±0.0190 | 0.4787±0.0291 | 0.0806±0.0232 | 0.4363±0.0213 | 0.1279±0.0304 | 0.4913±0.0364 | 0.1043±0.0250 | 0.2993±0.0221 | 0.1446±0.0288 | 0.4881±0.0345 |
| Probe + WV 4 | 0.1655±0.0005 | 1.1793±0.0029 | 0.2457±0.0007 | 1.2096±0.0066 | 0.2633±0.0008 | 1.4417±0.0034 | 0.2703±0.0009 | 1.1423±0.0054 | 0.3313±0.0003 | 1.4971±0.0116 |
| MSV 1 + WV 4 | 0.1654±0.0004 | 1.1819±0.0031 | 0.2461±0.0003 | 1.2106±0.0058 | 0.2641±0.0004 | 1.4449±0.0056 | 0.2714±0.0006 | 1.1412±0.0020 | 0.3321±0.0003 | 1.4995±0.0094 |
| MSV 4 | **0.1065**±0.0177 | **0.4677**±0.0362 | **0.0821**±0.0267 | **0.3663**±0.0206 | **0.1306**±0.0357 | **0.4793**±0.0424 | **0.0908**±0.0374 | **0.2472**±0.0384 | **0.1265**±0.0309 | **0.4424**±0.0423 |
| Probe + WV 16 | 0.1293±0.0009 | 0.9282±0.0023 | 0.1644±0.0015 | 0.6382±0.0023 | 0.1800±0.0017 | 0.8120±0.0069 | 0.1579±0.0032 | 0.5808±0.0108 | 0.2256±0.0010 | 0.8281±0.0047 |
| MSV 1 + WV 16 | 0.1290±0.0007 | 0.9299±0.0029 | 0.1655±0.0005 | 0.6443±0.0023 | 0.1824±0.0010 | 0.8198±0.0064 | 0.1611±0.0012 | 0.5922±0.0033 | 0.2275±0.0005 | 0.8366±0.0053 |
| MSV 16 | **0.1127**±0.0108 | **0.4615**±0.0291 | **0.0869**±0.0197 | **0.3289**±0.0200 | **0.1317**±0.0197 | **0.4724**±0.0414 | **0.0844**±0.0234 | **0.2079**±0.0290 | **0.1268**±0.0205 | **0.4385**±0.0411 |
| Probe + WV 64 | 0.1205±0.0011 | 0.7788±0.0038 | 0.1437±0.0018 | 0.4894±0.0039 | 0.1611±0.0026 | 0.6564±0.0086 | 0.1288±0.0037 | 0.4265±0.0148 | 0.1999±0.0014 | 0.6564±0.0032 |
| MSV 1 + WV 64 | **0.1202**±0.0008 | 0.7732±0.0032 | 0.1451±0.0007 | 0.5024±0.0022 | 0.1664±0.0034 | 0.6646±0.0065 | 0.1347±0.0039 | 0.4444±0.0042 | 0.2020±0.0007 | 0.6662±0.0044 |
| MSV 64 | 0.1204±0.0109 | **0.5443**±0.0501 | **0.0556**±0.0135 | **0.2904**±0.0230 | **0.1085**±0.0122 | **0.4986**±0.0234 | **0.0350**±0.0166 | **0.1501**±0.0213 | **0.0867**±0.0106 | **0.3705**±0.0394 |

**Table 25:** Best-of-$N$ accuracy, with verifiers at $N = 1, 4, 16, 64$.

| $N$ | Method | MATH | OlympiadBench | AMC12 | AIME | Omni-MATH |
|---|---|---|---|---|---|---|
| 1 | Probe | 0.6755±0.0000 | 0.4105±0.0000 | 0.3965±0.0000 | 0.2748±0.0000 | 0.2432±0.0000 |
| | MSV 1 | 0.6755±0.0000 | 0.4105±0.0000 | 0.3965±0.0000 | 0.2748±0.0000 | 0.2432±0.0000 |
| 4 | Probe | 0.7197±0.0006 | 0.4725±0.0008 | **0.4861**±0.0041 | 0.3771±0.0012 | 0.2885±0.0005 |
| | MSV 1 | 0.7155±0.0024 | 0.4681±0.0011 | 0.4854±0.0043 | 0.3758±0.0008 | 0.2878±0.0018 |
| | Probe + WV 4 | **0.7250**±0.0009 | **0.4770**±0.0003 | 0.4856±0.0030 | 0.3847±0.0010 | **0.2910**±0.0009 |
| | MSV 1 + WV 4 | 0.7224±0.0011 | 0.4755±0.0014 | 0.4845±0.0030 | **0.3852**±0.0008 | 0.2900±0.0012 |
| | MSV 4 | 0.7219±0.0012 | 0.4747±0.0021 | 0.4837±0.0020 | 0.3811±0.0018 | 0.2891±0.0015 |
| 16 | Probe | 0.7345±0.0014 | 0.5039±0.0023 | 0.5198±0.0037 | 0.4470±0.0025 | 0.3176±0.0020 |
| | MSV 1 | 0.7238±0.0031 | 0.4895±0.0037 | 0.5231±0.0035 | 0.4325±0.0050 | 0.3102±0.0023 |
| | Probe + WV 16 | 0.7450±0.0028 | 0.5087±0.0019 | 0.5254±0.0072 | 0.4481±0.0096 | 0.3162±0.0037 |
| | MSV 1 + WV 16 | 0.7440±0.0025 | 0.5104±0.0008 | 0.5235±0.0055 | 0.4460±0.0076 | 0.3133±0.0039 |
| | MSV 16 | **0.7472**±0.0009 | **0.5154**±0.0037 | **0.5399**±0.0061 | **0.4615**±0.0046 | **0.3198**±0.0032 |
| 64 | Probe | 0.7429±0.0030 | 0.5112±0.0062 | 0.5269±0.0154 | 0.4897±0.0098 | 0.3335±0.0077 |
| | MSV 1 | 0.7246±0.0058 | 0.4629±0.0110 | 0.5164±0.0119 | 0.4621±0.0140 | 0.3179±0.0061 |
| | Probe + WV 64 | 0.7500±0.0032 | 0.5223±0.0058 | 0.5388±0.0073 | 0.4790±0.0067 | 0.3210±0.0017 |
| | MSV 1 + WV 64 | 0.7442±0.0033 | 0.5210±0.0030 | 0.5313±0.0087 | 0.4714±0.0052 | 0.3179±0.0023 |
| | MSV 64 | **0.7607**±0.0033 | **0.5366**±0.0064 | **0.5866**±0.0138 | **0.5098**±0.0073 | **0.3540**±0.0023 |

**Table 26:** ECE and Brier score of confidence output on best-of-$N$ answers, with verifiers at $N = 1, 4, 16, 64$.

| | | MATH | | OlympiadBench | | AMC12 | | AIME | | Omni-MATH | |
|---|---|---|---|---|---|---|---|---|---|---|---|
| $N$ | Method | ECE ↓ | Brier ↓ | ECE ↓ | Brier ↓ | ECE ↓ | Brier ↓ | ECE ↓ | Brier ↓ | ECE ↓ | Brier ↓ |
| 1 | Probe | 0.0726±0.0401 | 0.2693±0.0214 | 0.0700±0.0517 | 0.2543±0.0288 | 0.1064±0.0581 | 0.2876±0.0429 | 0.1024±0.0483 | 0.1717±0.0295 | 0.1206±0.0589 | 0.2681±0.0504 |
| | MSV 1 | 0.0849±0.0190 | 0.2920±0.0122 | 0.0806±0.0232 | 0.2620±0.0122 | 0.1279±0.0304 | 0.2920±0.0159 | 0.1043±0.0250 | 0.1638±0.0123 | 0.1446±0.0288 | 0.2920±0.0206 |
| 4 | Probe | 0.1168±0.0448 | 0.3001±0.0341 | 0.1510±0.0689 | 0.3376±0.0653 | 0.1801±0.0639 | 0.3806±0.0666 | 0.1703±0.0579 | 0.2587±0.0594 | 0.2298±0.0777 | 0.3997±0.0995 |
| | MSV 1 | 0.1316±0.0206 | 0.3316±0.0176 | 0.1675±0.0276 | 0.3624±0.0257 | 0.1985±0.0293 | 0.3927±0.0311 | 0.1661±0.0269 | 0.2459±0.0242 | 0.2514±0.0319 | 0.4372±0.0374 |
| | Probe + WV 4 | 0.1797±0.0039 | 0.3856±0.0043 | 0.3066±0.0089 | 0.5660±0.0129 | 0.3050±0.0073 | 0.6019±0.0122 | 0.3204±0.0133 | 0.5777±0.0180 | 0.4396±0.0109 | 0.7359±0.0210 |
| | MSV 1 + WV 4 | 0.1815±0.0034 | 0.3871±0.0036 | 0.3044±0.0066 | 0.5642±0.0105 | 0.3014±0.0063 | 0.5945±0.0065 | 0.3149±0.0079 | 0.5680±0.0066 | 0.4354±0.0091 | 0.7306±0.0156 |
| | MSV 4 | **0.1133**±0.0187 | **0.3010**±0.0118 | **0.1031**±0.0345 | **0.2864**±0.0178 | **0.1477**±0.0412 | **0.3477**±0.0268 | **0.1019**±0.0489 | **0.2040**±0.0319 | **0.1695**±0.0402 | **0.3410**±0.0243 |
| 16 | Probe | 0.1559±0.0369 | 0.3397±0.0415 | 0.2294±0.0679 | 0.4428±0.0920 | 0.2657±0.0607 | 0.5024±0.0907 | 0.2364±0.0601 | 0.3776±0.0858 | 0.3296±0.0823 | 0.5635±0.1360 |
| | MSV 1 | 0.1740±0.0172 | 0.3818±0.0229 | 0.2519±0.0267 | 0.4830±0.0365 | 0.2725±0.0268 | 0.5147±0.0381 | 0.2407±0.0280 | 0.3754±0.0442 | 0.3542±0.0327 | 0.6170±0.0501 |
| | Probe + WV 16 | 0.1353±0.0017 | 0.3407±0.0023 | 0.2073±0.0070 | 0.4405±0.0035 | 0.1988±0.0046 | 0.4744±0.0030 | 0.1633±0.0035 | 0.4286±0.0055 | 0.3195±0.0064 | 0.5521±0.0111 |
| | MSV 1 + WV 16 | 0.1356±0.0007 | 0.3452±0.0024 | 0.2049±0.0053 | 0.4476±0.0042 | 0.2002±0.0019 | 0.4760±0.0014 | 0.1659±0.0040 | 0.4352±0.0036 | 0.3201±0.0056 | 0.5540±0.0093 |
| | MSV 16 | **0.1160**±0.0142 | **0.2905**±0.0079 | **0.1059**±0.0288 | **0.2859**±0.0204 | **0.1462**±0.0273 | **0.3516**±0.0201 | **0.0876**±0.0301 | **0.2067**±0.0236 | **0.1776**±0.0328 | **0.3583**±0.0310 |
| 64 | Probe | 0.1848±0.0276 | 0.3802±0.0400 | 0.3048±0.0552 | 0.5520±0.0995 | 0.3288±0.0573 | 0.6132±0.1003 | 0.3013±0.0588 | 0.5159±0.1006 | 0.4185±0.0796 | 0.7348±0.1555 |
| | MSV 1 | 0.2128±0.0143 | 0.4419±0.0240 | 0.3555±0.0312 | 0.6453±0.0520 | 0.3520±0.0174 | 0.6636±0.0295 | 0.3162±0.0348 | 0.5324±0.0653 | 0.4504±0.0303 | 0.8118±0.0537 |
| | Probe + WV 64 | 0.1259±0.0029 | 0.3305±0.0019 | 0.1736±0.0020 | 0.4102±0.0041 | 0.1823±0.0155 | 0.4611±0.0027 | 0.1222±0.0168 | 0.4165±0.0138 | 0.2888±0.0049 | 0.5115±0.0042 |
| | MSV 1 + WV 64 | 0.1300±0.0016 | 0.3329±0.0017 | 0.1769±0.0059 | 0.4219±0.0016 | 0.1929±0.0086 | 0.4586±0.0038 | 0.1302±0.0100 | 0.4232±0.0069 | 0.2920±0.0033 | 0.5175±0.0060 |
| | MSV 64 | **0.1323**±0.0147 | **0.2974**±0.0080 | **0.0803**±0.0281 | **0.2613**±0.0315 | **0.1731**±0.0207 | **0.4103**±0.0187 | **0.0749**±0.0500 | **0.2351**±0.0542 | **0.1303**±0.0273 | **0.3434**±0.0401 |

**Table 27:** AUROC and Brier score of verifiers trained and evaluated on streaming answers.

| | MATH | | OlympiadBench | | AMC12 | | AIME | | Omni-MATH | |
|---|---|---|---|---|---|---|---|---|---|---|
| | AUROC ↑ | Brier ↓ | AUROC ↑ | Brier ↓ | AUROC ↑ | Brier ↓ | AUROC ↑ | Brier ↓ | AUROC ↑ | Brier ↓ |
| Probe | 0.8690±0.0135 | 0.1592±0.0101 | 0.8317±0.0089 | 0.1478±0.0232 | 0.8621±0.0108 | 0.1413±0.0195 | 0.9114±0.0142 | 0.0796±0.0105 | 0.8229±0.0142 | 0.1527±0.0242 |
| MSV 1 | 0.8443±0.0223 | 0.1601±0.0125 | 0.8169±0.0135 | 0.1425±0.0076 | 0.8546±0.0104 | 0.1343±0.0028 | 0.9056±0.0076 | 0.0642±0.0028 | 0.7951±0.0125 | 0.1381±0.0084 |
| Probe + WV 4 | 0.8769±0.0058 | 0.1563±0.0079 | 0.8186±0.0121 | 0.1493±0.0183 | 0.8561±0.0069 | 0.1381±0.0149 | 0.8740±0.0177 | 0.0768±0.0069 | 0.8167±0.0049 | 0.1246±0.0197 |
| MSV 1 + WV 4 | 0.8567±0.0130 | 0.1629±0.0074 | 0.8148±0.0113 | 0.1435±0.0113 | 0.8556±0.0067 | 0.1310±0.0074 | 0.9055±0.0075 | 0.0657±0.0028 | 0.7813±0.0039 | 0.1293±0.0082 |
| MSV 4 | **0.8917**±0.0133 | **0.1324**±0.0034 | **0.8683**±0.0169 | **0.1197**±0.0094 | **0.9002**±0.0118 | **0.1068**±0.0033 | **0.9312**±0.0101 | **0.0472**±0.0053 | **0.8327**±0.0127 | **0.1160**±0.0088 |
| Probe + WV 16 | 0.8769±0.0058 | 0.1563±0.0079 | 0.8186±0.0121 | 0.1493±0.0183 | 0.8561±0.0069 | 0.1381±0.0149 | 0.8740±0.0177 | 0.0768±0.0069 | 0.8167±0.0049 | 0.1246±0.0197 |
| MSV 1 + WV 16 | 0.8567±0.0130 | 0.1629±0.0074 | 0.8148±0.0113 | 0.1435±0.0113 | 0.8556±0.0067 | 0.1310±0.0074 | 0.9055±0.0075 | 0.0657±0.0028 | 0.7813±0.0039 | 0.1293±0.0082 |
| MSV 16 | **0.9037**±0.0201 | **0.1104**±0.0105 | **0.8946**±0.0229 | **0.1054**±0.0076 | **0.9105**±0.0115 | **0.1041**±0.0049 | **0.9532**±0.0069 | **0.0375**±0.0029 | **0.8591**±0.0029 | **0.1100**±0.0063 |
| Probe + WV 64 | 0.8769±0.0058 | 0.1563±0.0079 | 0.8186±0.0121 | 0.1493±0.0183 | 0.8561±0.0069 | 0.1381±0.0149 | 0.8740±0.0177 | 0.0768±0.0069 | 0.8167±0.0049 | 0.1246±0.0197 |
| MSV 1 + WV 64 | 0.8567±0.0130 | 0.1629±0.0074 | 0.8148±0.0113 | 0.1435±0.0074 | 0.8556±0.0067 | 0.1310±0.0074 | 0.9055±0.0075 | 0.0657±0.0028 | 0.7813±0.0039 | 0.1293±0.0082 |
| MSV 64 | **0.9128**±0.0125 | **0.1147**±0.0022 | **0.8879**±0.0136 | **0.1072**±0.0118 | **0.8953**±0.0034 | **0.1173**±0.0033 | **0.9648**±0.0016 | **0.0323**±0.0013 | **0.8352**±0.0084 | **0.1255**±0.0051 |

**Table 28:** ECE and NLL of verifiers trained and evaluated on streaming answers.

| | MATH | | OlympiadBench | | AMC12 | | AIME | | Omni-MATH | |
|---|---|---|---|---|---|---|---|---|---|---|
| | ECE ↓ | NLL ↓ | ECE ↓ | NLL ↓ | ECE ↓ | NLL ↓ | ECE ↓ | NLL ↓ | ECE ↓ | NLL ↓ |
| Probe | 0.0934±0.0416 | 0.5184±0.0528 | 0.1317±0.0382 | 0.5206±0.0914 | 0.1292±0.0461 | 0.4841±0.0689 | 0.0901±0.0384 | 0.2781±0.0415 | 0.1656±0.0470 | 0.5194±0.0945 |
| MSV 1 | 0.0939±0.0163 | 0.5356±0.0441 | 0.0911±0.0163 | 0.5055±0.0208 | 0.0862±0.0164 | 0.4532±0.0039 | 0.0511±0.0229 | 0.2412±0.0113 | 0.1157±0.0132 | 0.4974±0.0232 |
| Probe + WV 4 | 0.1052±0.0353 | 0.4842±0.0382 | 0.1817±0.0347 | 0.4885±0.0544 | 0.1272±0.0413 | 0.4410±0.0489 | 0.0861±0.0425 | 0.2784±0.0348 | 0.1104±0.0381 | 0.4407±0.0664 |
| MSV 1 + WV 4 | 0.0822±0.0132 | 0.5204±0.0321 | 0.1292±0.0200 | 0.4836±0.0315 | 0.0767±0.0141 | 0.4272±0.0175 | 0.0548±0.0249 | 0.2447±0.0113 | 0.1030±0.0152 | 0.4550±0.0223 |
| MSV 4 | **0.0814**±0.0133 | **0.4637**±0.0294 | **0.0797**±0.0143 | **0.4289**±0.0270 | **0.0638**±0.0093 | **0.3709**±0.0172 | **0.0271**±0.0120 | **0.1814**±0.0187 | **0.0959**±0.0116 | **0.4206**±0.0287 |
| Probe + WV 16 | 0.1052±0.0353 | 0.4842±0.0382 | 0.1817±0.0347 | 0.4885±0.0544 | 0.1272±0.0413 | 0.4410±0.0489 | 0.0861±0.0425 | 0.2784±0.0348 | 0.1104±0.0381 | 0.4407±0.0664 |
| MSV 1 + WV 16 | 0.0822±0.0132 | 0.5204±0.0321 | 0.1292±0.0200 | 0.4836±0.0315 | 0.0767±0.0141 | 0.4272±0.0175 | 0.0548±0.0249 | 0.2447±0.0113 | 0.1030±0.0152 | 0.4550±0.0223 |
| MSV 16 | **0.0666**±0.0211 | **0.3781**±0.0461 | **0.0606**±0.0166 | **0.3546**±0.0195 | **0.0563**±0.0108 | **0.3672**±0.0173 | **0.0333**±0.0146 | **0.1498**±0.0109 | **0.0883**±0.0122 | **0.3888**±0.0163 |
| Probe + WV 64 | 0.1052±0.0353 | 0.4842±0.0382 | 0.1817±0.0347 | 0.4885±0.0544 | 0.1272±0.0413 | 0.4410±0.0489 | 0.0861±0.0425 | 0.2784±0.0348 | 0.1104±0.0381 | 0.4407±0.0664 |
| MSV 1 + WV 64 | **0.0822**±0.0132 | 0.5204±0.0321 | 0.1292±0.0200 | 0.4836±0.0315 | **0.0767**±0.0141 | 0.4272±0.0175 | 0.0548±0.0249 | 0.2447±0.0113 | **0.1030**±0.0152 | 0.4550±0.0223 |
| MSV 64 | 0.0913±0.0151 | **0.4135**±0.0322 | **0.0868**±0.0236 | **0.3770**±0.0346 | 0.1054±0.0065 | 0.4293±0.0210 | **0.0165**±0.0048 | **0.1254**±0.0027 | 0.1316±0.0069 | **0.4296**±0.0284 |

**Table 29:** Area Under the Tradeoff Curve (AUTC) of verifiers trained and evaluated on streaming answers.

| | MATH | OlympiadBench | AMC12 | AIME | Omni-MATH |
|---|---|---|---|---|---|
| Probe | 1515.60±4.54 | 1018.33±5.09 | 924.22±13.90 | 563.44±19.34 | 651.90±6.05 |
| MSV 1 | 1508.67±4.09 | 1007.82±7.29 | 928.82±10.99 | 551.56±7.02 | 652.47±3.12 |
| Probe + WV 4 | 2074.56±11.23 | 798.03±13.92 | 1232.06±17.58 | 808.66±38.60 | 790.69±7.91 |
| MSV 1 + WV 4 | 2034.65±5.09 | 1220.65±14.54 | 1247.01±14.53 | 789.28±17.55 | 780.09±6.47 |
| MSV 4 | **2118.07**±8.79 | **1282.20**±19.29 | **1297.64**±16.58 | **850.44**±8.78 | **827.07**±11.08 |
| Probe + WV 16 | 2387.20±22.08 | 1395.88±17.53 | 1412.28±20.37 | 1000.25±32.17 | 899.39±9.63 |
| MSV 1 + WV 16 | 2280.68±17.58 | 1286.12±20.30 | 1432.93±27.27 | 959.76±28.52 | 860.55±6.27 |
| MSV 16 | **2473.59**±12.96 | **1460.66**±22.79 | **1475.91**±21.36 | **1078.29**±18.00 | **948.81**±10.43 |
| Probe + WV 64 | 2554.70±8.52 | 1464.85±22.84 | 1503.08±27.02 | 1167.81±40.73 | 973.23±12.37 |
| MSV 1 + WV 64 | 2347.04±26.68 | 1287.00±22.95 | 1435.32±33.72 | 1045.05±40.58 | 903.09±20.84 |
| MSV 64 | **2677.20**±22.41 | **1551.33**±46.96 | **1606.40**±32.74 | **1289.80**±15.48 | **1046.36**±20.24 |

**Table 30:** AUROC and Brier score of training-free baseline methods for terminal answers.

| | MATH | | OlympiadBench | | AMC12 | | AIME | | Omni-MATH | |
|---|---|---|---|---|---|---|---|---|---|---|
| | AUROC ↑ | Brier ↓ | AUROC ↑ | Brier ↓ | AUROC ↑ | Brier ↓ | AUROC ↑ | Brier ↓ | AUROC ↑ | Brier ↓ |
| Token Probs | 0.5320 | 0.2504 | 0.7704 | 0.3696 | 0.7350 | 0.3540 | 0.8999 | 0.3073 | 0.7814 | 0.4467 |
| Token Probs | 0.5320 | 0.2504 | 0.7704 | 0.3696 | 0.7350 | 0.3540 | 0.8999 | 0.3073 | 0.7814 | 0.4467 |
| Token Probs + WV 4 | **0.7922** | **0.1678** | **0.8392** | **0.2166** | **0.7978** | **0.2448** | **0.8223** | **0.2239** | **0.8085** | **0.2620** |
| Self-consistency 4 | 0.7876 | 0.1686 | 0.8290 | 0.2180 | 0.7893 | 0.2477 | 0.8001 | 0.2274 | 0.7986 | 0.2627 |
| Token Probs | 0.5320 | 0.2504 | 0.7704 | 0.3696 | 0.7350 | 0.3540 | 0.8999 | 0.3073 | 0.7814 | 0.4467 |
| Token Probs + WV 16 | **0.8155** | **0.1474** | **0.8801** | **0.1690** | **0.8364** | **0.1961** | **0.8531** | **0.1662** | **0.8457** | **0.1968** |
| Self-consistency 16 | 0.8149 | 0.1482 | 0.8751 | 0.1704 | 0.8299 | 0.1991 | 0.8422 | 0.1696 | 0.8398 | 0.1976 |
| Token Probs | 0.5320 | 0.2504 | 0.7704 | 0.3696 | 0.7350 | 0.3540 | 0.8999 | 0.3073 | 0.7814 | 0.4467 |
| Token Probs + WV 64 | **0.8304** | **0.1432** | **0.8901** | **0.1591** | **0.8454** | **0.1871** | **0.8608** | **0.1549** | **0.8531** | **0.1841** |
| Self-consistency 64 | 0.8273 | 0.1440 | 0.8875 | 0.1605 | 0.8410 | 0.1900 | 0.8517 | 0.1583 | 0.8505 | 0.1849 |

**Table 31:** ECE and NLL of training-free baseline methods for terminal answers.

| | MATH | | OlympiadBench | | AMC12 | | AIME | | Omni-MATH | |
|---|---|---|---|---|---|---|---|---|---|---|
| | ECE ↓ | NLL ↓ | ECE ↓ | NLL ↓ | ECE ↓ | NLL ↓ | ECE ↓ | NLL ↓ | ECE ↓ | NLL ↓ |
| Token Probs | 0.1855 | 0.9190 | 0.4124 | 1.1444 | 0.3935 | 1.1384 | 0.4198 | 0.8669 | 0.5389 | 1.2505 |
| Token Probs + WV 4 | 0.1615 | **1.1805** | 0.2416 | **1.2018** | 0.2616 | **1.4551** | 0.2677 | **1.1803** | 0.3306 | **1.4633** |
| Self-consistency 4 | **0.1611** | 1.1823 | **0.2408** | 1.2054 | **0.2601** | 1.4620 | **0.2664** | 1.1889 | **0.3300** | 1.4647 |
| Token Probs + WV 16 | 0.1212 | **0.9249** | 0.1566 | **0.6566** | 0.1752 | **0.8604** | 0.1545 | **0.6618** | 0.2235 | **0.8289** |
| Self-consistency 16 | **0.1204** | 0.9264 | **0.1551** | 0.6605 | **0.1737** | 0.8673 | **0.1517** | 0.6702 | **0.2226** | 0.8306 |
| Token Probs + WV 64 | 0.1158 | **0.7592** | 0.1346 | **0.5194** | 0.1549 | **0.7096** | 0.1254 | **0.5219** | 0.1970 | **0.6686** |
| Self-consistency 64 | **0.1149** | 0.7598 | **0.1329** | 0.5231 | **0.1531** | 0.7161 | **0.1221** | 0.5301 | **0.1959** | 0.6692 |

**Table 32:** Best-of-$N$ accuracy (ACC) with training-free baseline methods at $N = 1, 4, 16, 64$.

| $N$ | Method | MATH | OlympiadBench | AMC12 | AIME | Omni-MATH |
|---|---|---|---|---|---|---|
| 1 | Token Probs | 0.6755 | 0.4105 | 0.3965 | 0.2748 | 0.2432 |
| 4 | Token Probs | 0.6943 | 0.4495 | 0.4445 | 0.3313 | 0.2667 |
| | Token Probs + WV 4 | **0.7102** | **0.4590** | **0.4450** | **0.3410** | **0.2723** |
| | Self-consistency 4 | 0.7049 | 0.4484 | 0.4286 | 0.3251 | 0.2633 |
| 16 | Token Probs | 0.6959 | 0.4754 | 0.4664 | **0.3772** | **0.2835** |
| | Token Probs + WV 16 | **0.7221** | **0.4788** | **0.4683** | 0.3711 | **0.2835** |
| | Self-consistency 16 | 0.7210 | 0.4676 | 0.4478 | 0.3566 | 0.2801 |
| 64 | Token Probs | 0.7098 | **0.4978** | **0.5224** | **0.4129** | **0.3013** |
| | Token Probs + WV 64 | 0.7299 | 0.4888 | 0.4627 | 0.3728 | 0.2902 |
| | Self-consistency 64 | **0.7321** | 0.4844 | 0.4403 | 0.3594 | 0.2790 |

**Table 33:** ECE and Brier score of confidence output on best-of $N$ answers with training-free baseline methods at $N = 1, 4, 16, 64$.

| $N$ | Method | MATH ECE↓ | MATH Brier↓ | OlympiadBench ECE↓ | OlympiadBench Brier↓ | AMC12 ECE↓ | AMC12 Brier↓ | AIME ECE↓ | AIME Brier↓ | Omni-MATH ECE↓ | Omni-MATH Brier↓ |
|---|---|---|---|---|---|---|---|---|---|---|---|
| 1 | Token Probs | 0.1855 | 0.5007 | 0.4124 | 0.7393 | 0.3935 | 0.7081 | 0.4198 | 0.6146 | 0.5389 | 0.8933 |
| 4 | Token Probs | 0.2157 | 0.5126 | 0.4320 | 0.7905 | 0.4109 | 0.7549 | 0.4447 | 0.6956 | 0.5775 | 1.0020 |
| | Token Probs + WV 4 | **0.1588** | 0.3661 | **0.2549** | 0.5065 | **0.2763** | 0.5557 | **0.2789** | 0.5421 | **0.3698** | 0.6127 |
| | Self-consistency 4 | 0.1613 | **0.3638** | 0.2586 | **0.4983** | 0.2843 | **0.5504** | 0.2805 | **0.5316** | 0.3708 | **0.6031** |
| 16 | Token Probs | 0.2517 | 0.5414 | 0.4480 | 0.8347 | 0.4372 | 0.8027 | 0.4599 | 0.7566 | 0.6065 | 1.0912 |
| | Token Probs + WV 16 | 0.1296 | **0.3381** | **0.1907** | 0.4404 | **0.2094** | 0.4898 | **0.1849** | 0.4595 | **0.2952** | 0.5031 |
| | Self-consistency 16 | **0.1292** | 0.3404 | 0.1971 | **0.4347** | 0.2235 | **0.4840** | 0.1927 | **0.4558** | 0.2961 | **0.5021** |
| 64 | Token Probs | 0.2653 | 0.5416 | 0.4556 | 0.8656 | 0.4143 | 0.7896 | 0.4757 | 0.8202 | 0.6271 | 1.1669 |
| | Token Probs + WV 64 | 0.1305 | **0.3318** | **0.1670** | 0.4248 | **0.2075** | 0.4640 | **0.1663** | 0.4364 | **0.2742** | 0.4855 |
| | Self-consistency 64 | **0.1259** | 0.3377 | 0.1680 | 0.4279 | 0.2196 | **0.4502** | 0.1733 | **0.4341** | 0.2830 | **0.4749** |

**Table 34:** AUROC and Brier score of training-free baseline methods on streaming answers.

| | MATH AUROC↑ | MATH Brier↓ | OlympiadBench AUROC↑ | OlympiadBench Brier↓ | AMC12 AUROC↑ | AMC12 Brier↓ | AIME AUROC↑ | AIME Brier↓ | Omni-MATH AUROC↑ | Omni-MATH Brier↓ |
|---|---|---|---|---|---|---|---|---|---|---|
| Token Probs | 0.7910 | 0.3399 | 0.8091 | 0.4266 | 0.7856 | 0.3979 | 0.8858 | 0.3217 | 0.7974 | 0.4746 |
| Token Probs | **0.7910** | 0.3399 | **0.8091** | 0.4266 | **0.7856** | 0.3979 | **0.8858** | 0.3217 | **0.7974** | 0.4746 |
| Token Probs + WV 4 | 0.7867 | 0.3341 | 0.8028 | 0.4149 | 0.7774 | 0.3892 | 0.8784 | 0.3107 | 0.7889 | 0.4609 |
| Self-consistency 4 | 0.5000 | **0.2500** | 0.5000 | **0.2500** | 0.5000 | **0.2500** | 0.5000 | **0.2500** | 0.5000 | **0.2500** |
| Token Probs | **0.7910** | 0.3399 | **0.8091** | 0.4266 | **0.7856** | 0.3979 | **0.8858** | 0.3217 | **0.7974** | 0.4746 |
| Token Probs + WV 16 | 0.7867 | 0.3341 | 0.8028 | 0.4149 | 0.7774 | 0.3892 | 0.8784 | 0.3107 | 0.7889 | 0.4609 |
| Self-consistency 16 | 0.5000 | **0.2500** | 0.5000 | **0.2500** | 0.5000 | **0.2500** | 0.5000 | **0.2500** | 0.5000 | **0.2500** |
| Token Probs | **0.7910** | 0.3399 | **0.8091** | 0.4266 | **0.7856** | 0.3979 | **0.8858** | 0.3217 | **0.7974** | 0.4746 |
| Token Probs + WV 64 | 0.7867 | 0.3341 | 0.8028 | 0.4149 | 0.7774 | 0.3892 | 0.8784 | 0.3107 | 0.7889 | 0.4609 |
| Self-consistency 64 | 0.5000 | **0.2500** | 0.5000 | **0.2500** | 0.5000 | **0.2500** | 0.5000 | **0.2500** | 0.5000 | **0.2500** |

**Table 35:** ECE and NLL of training-free baseline methods on streaming answers.

| | MATH ECE↓ | MATH NLL↓ | OlympiadBench ECE↓ | OlympiadBench NLL↓ | AMC12 ECE↓ | AMC12 NLL↓ | AIME ECE↓ | AIME NLL↓ | Omni-MATH ECE↓ | Omni-MATH NLL↓ |
|---|---|---|---|---|---|---|---|---|---|---|
| Token Probs | 0.3732 | 1.1043 | 0.5255 | 1.1625 | 0.4944 | 1.1466 | 0.4886 | 0.8724 | 0.6045 | 1.2632 |
| Token Probs | 0.3732 | 1.1043 | 0.5255 | 1.1625 | 0.4944 | 1.1466 | 0.4886 | 0.8724 | 0.6045 | 1.2632 |
| Token Probs + WV 4 | 0.3644 | 1.0481 | 0.5135 | 1.1115 | 0.4840 | 1.0962 | 0.4757 | 0.8341 | 0.5929 | 1.2095 |
| Self-consistency 4 | **0.0226** | **0.6931** | **0.2553** | **0.6931** | **0.2529** | **0.6931** | **0.3683** | **0.6931** | **0.3533** | **0.6931** |
| Token Probs | 0.3732 | 1.1043 | 0.5255 | 1.1625 | 0.4944 | 1.1466 | 0.4886 | 0.8724 | 0.6045 | 1.2632 |
| Token Probs + WV 16 | 0.3644 | 1.0481 | 0.5135 | 1.1115 | 0.4840 | 1.0962 | 0.4757 | 0.8341 | 0.5929 | 1.2095 |
| Self-consistency 16 | **0.0226** | **0.6931** | **0.2553** | **0.6931** | **0.2529** | **0.6931** | **0.3683** | **0.6931** | **0.3533** | **0.6931** |
| Token Probs | 0.3732 | 1.1043 | 0.5255 | 1.1625 | 0.4944 | 1.1466 | 0.4886 | 0.8724 | 0.6045 | 1.2632 |
| Token Probs + WV 64 | 0.3644 | 1.0481 | 0.5135 | 1.1115 | 0.4840 | 1.0962 | 0.4757 | 0.8341 | 0.5929 | 1.2095 |
| Self-consistency 64 | **0.0226** | **0.6931** | **0.2553** | **0.6931** | **0.2529** | **0.6931** | **0.3683** | **0.6931** | **0.3533** | **0.6931** |

**Table 36:** Area Under the Tradeoff Curve (AUTC) of training-free baseline methods.

|  | MATH | OlympiadBench | AMC12 | AIME | Omni-MATH |
|---|---|---|---|---|---|
| Token Probs | 1246.1625 | 874.4062 | 780.8859 | 423.6729 | 597.4429 |
| Token Probs | 1315.0728 | 804.3277 | 706.4964 | 353.0816 | 574.9029 |
| Token Probs + WV 4 | 1341.3023 | 793.8698 | 721.3110 | 354.8151 | 572.9540 |
| Self-consistency 4 | **1560.7934** | **944.5491** | **861.8922** | **550.2739** | **655.1951** |
| Token Probs | 1085.6131 | 712.8956 | 629.6273 | 322.1713 | 551.7789 |
| Token Probs + WV 16 | 1120.1693 | 699.5992 | 649.8307 | 358.4021 | 555.1394 |
| Self-consistency 16 | **1628.0505** | **960.6178** | **879.9306** | **565.9113** | **660.7702** |
| Token Probs | 836.6923 | 665.0830 | 557.9867 | 298.3862 | 532.7231 |
| Token Probs + WV 64 | 869.1815 | 670.9667 | 609.4350 | 321.8818 | 526.9107 |
| Self-consistency 64 | **1665.4385** | **998.9186** | **925.0327** | **545.8705** | **652.2426** |

