# OpenReview forum: "Parallel Test-Time Scaling with Multi-Sequence Verifiers"
_ICLR.cc/2026/Conference — Submitted to ICLR 2026_

### Official Review · Reviewer_cGUP · 2025-10-26

**Soundness:** 3
**Presentation:** 3
**Contribution:** 3
**Rating:** 6
**Confidence:** 4

**Summary:**

The paper proposes Multi-Sequence Verifier (MSV), a training algorithm to train a verifier model for test-time scaling, that can score multiple sequences in parallel, importantly taking into account all other sequences when computing a score for a sequence. This contrasts it from sequence-level methods like best-of-n with external verifiers.

**I think the paper should be accepted.** However, some crucial aspects should be addressed by the authors (see "Weaknesses" below), most importantly a more detailed latency breakdown, and comparing against other commonly used baselines (e.g. BoN with external verifier).

**Strengths:**

### Writing
The paper is very well written and well structured. All concepts are cleanly defined and explained (except a unifying overview of the method, see "Weaknesses" below).

### Contribution
The proposed method seems quite novel and interesting. It features various novel ideas (see section 3).

### Results
The results are look promising. Not only does MSV seem to lead to accuracy gains, but it also seems to lead to well-calibrated models, which is possibly independently relevant to the research community.

**Weaknesses:**

### Calibration
From a theoretical point of view, it's not clear (at least to me) that the proposed procedure of computing $\tilde{y}$ (which I'm assuming is used as the predictive probability $p$ in section 3.3) actually enforces calibration (even if the experiments show this happens empirically). Yet the authors make it seem like this naturally follows from the proposed training objective (e.g. in line 299). If this is indeed supported by theory, the authors should expand on it.

### Relation to Existing Literature
I believe some of the proposed setting in which MSV is trained, e.g. terminal answers vs streaming answers, the idea of extracting intermediate answers at delimiters, etc., is insufficiently put into context w.r.t. previous literature. If all of these ideas are novel, this should be made more clear, but I would assume that in particular the "terminal answers" vs "streaming answers" viewpoint has been studied before.

### Method
It would help to understand the method better if the authors provided an algorithm of how MSV is trained, that combines all the parts of section 3 and makes it easy to grasp what parts are being trained and how (as there are various trainable parameters scattered throughout section 3).

### Experimental Results
The proposed MSV contains various design choices that seem somewhat arbitrary, e.g. the choice of masks (section 3.2). Ablations over these masks and their affect on downstream performance would be helpful.

Furthermore, while the authors claim they show accuracy-latency tradeoffs, e.g. in Figure 7, they actually usually just show accuracy-token position tradeoffs. The only latency results I could find are in Table 4 (line 864 ff), and those are not very explicit. For example, how is "decoding" in this table defined? Is this only the latency for decoding tokens that would be generated as part of the sequences themselves, or does this include latency for decoding of the intermediate extracted answers? (Which should go into the latency overhead of MSV, but should be separated from the latency of BoN or other baselines.) The authors should include a more detailed latency comparison, that shows training time/compute, and a detailed breakdown of the latency overhead that MSV adds to simpler baselines.

The authors compare to various baselines in their experiments. However, a) some of these baselines are not explained at all (e.g. Probe + WV). Moreover, one baseline, MSV 1, is their own method in a single-sequence setup. Comparing to other, commonly used methods in the literature would make the experimental results much stronger. (E.g., how does this compare to simple BoN with an external verifier?)

**Questions:**

- minor: some of the fonts in figures and plots are too small to read (e.g. Figures 1, 4, and many of the tables)
- minor: line 895: "since forward pass" -> "since *a* forward pass"
- sometimes the authors write "multi-sample verifier" instead of "multi-sequence verifier" (caption of Figure 1, and line 59)
- line 158: "to end of $n$th sequence" -> "to *the* end..."
- minor notational inaccuracy: In the "Streaming Answers" setting, the authors should don't clarify which of the $K^{(n)}$ answers is the terminal one (I assume it's $a^{(n)}_{K^{(n)}}$, since the other answers are defined to correspond to the intermediate steps; this should be clarified though since saying that $a^{(n)}_1$ is the terminal answer in the terminal answers setting seems to contradict this notation)
- it would help to explain in more detail at what points in the generation intermediate answers are extracted in the streaming answers setting. The authors say "whenever a delimiter [...] is encountered". Could you say what delimiters the LLM you're using can produce exactly (any other than "wait"?), how often this happens, etc.?
- how do the "terminal answer" vs "streaming answer" viewpoints relate to existing literature?
- in the input concatenation (line 202), how do you concatenate exactly? Is there any delimiter/special structure by which you concatenate answers, or just raw concatenation? (Looking at the following section defining the masks, it is probably raw concatenation, but why this is reasonable only becomes clear in the following section, it might help to clarify this here.)
- the difference between "within-sequence mask" and "within-answer mask" could be made more clear, both in text as well as in Figure 1 where they look identical. (As I understand it, the former is all answers within that sequence up to a time $t$, while the latter is only one answer in that sequence, but it's not entirely clear.) In particular, there's some notation here (ans(u), seq(u), step(u)) that could be defined more clearly.
- the definition of the final feature (line 256) is not entirely clear to me. Why only take the last token's representation (and not a (weighted) mean across tokens)? Does the MLP simply map from $\mathbb{R}$ to $\mathbb{R}^d$?
- maybe I'm missing it, but it seems like the correctness probability $p$ (section 3.3) is not defined anywhere. Is this simply the predicted $\tilde{y}$?
- the training set seems to be very small (224 problems, cf. line 730). Could the authors elaborate on why the training set is chosen so small? How does the validation error look like during training, how long does training take, etc.? If MSV can be trained quickly, this could be highlighted as an advantage in the paper.
- the end of section 2.2 seems to be missing
- line 81: "... that latency doesn't grow" -> This claim doesn't seem to be supported anywhere (in particular, I was not able to find latency tradeoffs as $N$ grows). Also, if latency doesn't grow, scaling beyond $N=64$ would be interesting to see.

---

> ### Author Response · Authors · 2025-11-19
>
> We appreciate your thoughtful questions and feedback.
>
> ### W1. Calibration
>
> > From a theoretical point of view, it's not clear (at least to me) that the proposed procedure of computing $\tilde y$ (which I'm assuming is used as the predictive probability $p$ in section 3.3) actually enforces calibration
>
> The predictions $\tilde y$ are trained with cross-entropy, a strictly proper loss, which encourages calibrated probabilities under standard assumptions. [1] We will clarify this connection in the paper. And thank you for catching the typo, we replaced $p$ with $\tilde y$.
>
>
> ### W2. Relation to existing literature
>
> > I believe some of the proposed setting in which MSV is trained, e.g. terminal answers vs streaming answers, the idea of extracting intermediate answers at delimiters, etc., is insufficiently put into context w.r.t. previous literature.
>
> Indeed, the Streaming Answers setting with $N=1$ sequence has been explored by Zhang et al., 2025 and Yang et al., 2025, which we only briefly mention in the Introduction and Related Work sections. Both works extract intermediate answers at delimiters and perform early stopping by using some measure of confidence. In our revised manuscript, we emphasize this in Section 3.1, when introducing the Streaming Answers setting. In the contributions list in Introduction, we have also reworded “We introduce a novel parallel early-stopping framework,” to “We generalize an existing early-stopping framework to the parallel decoding setting,” so that readers are not misguided to thinking that we are also proposing the early stopping framework as a novel contribution.
>
> ### W3. Method
>
> > It would help to understand the method better if the authors provided an algorithm of how MSV is trained, that combines all the parts of section 3 and makes it easy to grasp what parts are being trained and how (as there are various trainable parameters scattered throughout section 3).
>
> https://anonymous.4open.science/api/repo/msv-rebuttal-iclr2026-016E/file/cGUP/algorithm.png
>
> We appreciate this suggestion, and totally agree that an algorithm pseudocode would help readers understand the method better. We have created the pseudocode for the MSV forward pass (See the image linked above) and have added it to section 3.2 in our revised manuscript.
>
> ### W4. Experimental Results
>
> > Ablations over these masks and their affect on downstream performance would be helpful.
>
> Thank you for raising this important point. We provide ablation over the masks in **Common Response 3**.
>
> ### W5. Latency
>
> > The authors should include a more detailed latency comparison, that shows training time/compute, and a detailed breakdown of the latency overhead that MSV adds to simpler baselines.
>
> We appreciate the concern about the lack of a detailed analysis of latency, which was an important missing part of our paper. We provide a detailed analysis of the latency for inference in **Common Response 2**.
>
> In the Terminal Answers setting, training takes around 50 minutes regardless of the verifier. In the Streaming Answers setting, training of Probe, MSV 1, MSV 4, MSV 16, and MSV 64 takes around 105, 105, 105, 110, and 145 minutes, respectively. We performed cross-validation on the number of epochs, meaning that running more epochs on Probe, MSV 1, or MSV 4 doesn’t improve their performance.
>
> ### W6. Baselines
>
> > some of these baselines are not explained at all (e.g. Probe + WV).
>
> Thank you for pointing this out. We didn’t explain weighted voting (WV) in the main text, and only explained it in Appendix A.1.2, which likely left the readers confused. We have added an explanation to it in Section 4.1, in our revised manuscript.
>
> > Comparing to other, commonly used methods in the literature would make the experimental results much stronger. (E.g., how does this compare to simple BoN with an external verifier?)
>
> __Table R.16__ Best-of-64 performance with Qwen2.5-Math-PRM-7B vs MSV 64
>
> | Dataset       | Qwen2.5-Math-PRM-7B | Qwen2.5-Math-PRM-7B + WV | MSV 64  |
> |---------------|---------------------:|-------------------------:|--------:|
> | MATH          | 0.7411               | 0.7433                   | **0.7607**  |
> | OlympiadBench | 0.3906               | 0.4018                   | **0.5366**  |
> | AMC12         | 0.5224               | 0.5000                   | **0.5866**  |
> | AIME          | 0.4397               | 0.4174                   | **0.5098**  |
> | Omni-MATH     | 0.2344               | 0.2366                   | **0.3540**  |
>
>
> We very much appreciate this suggestion. We performed experiments with Qwen2.5-Math-PRM-7B, [2] which is one of the most performant open-source verifiers as reported in [3]. Table R.16 reports the best-of-64 scores across all benchmarks.

---

> ### Author Response · Authors · 2025-11-19
>
> ### Responses to Questions
>
> We sincerely thank the reviewer for all the questions. These are extremely helpful in improving our paper.
>
> > Could you say what delimiters the LLM you're using can produce exactly (any other than "wait"?), how often this happens, etc.?
>
> We only use “Wait”, which occurs once every 200 tokens on average, on the AIME dataset. We also experimented with two other delimiters, “Alternatively” and “But”, which occur once every 900 and 400 tokens, respectively. Please see Table R.15 in our response to reviewer 5z5v for the results.
>
> > Is there any delimiter/special structure by which you concatenate answers, or just raw concatenation?
>
> It is indeed just raw concatenation.
>
> > Why only take the last token's representation (and not a (weighted) mean across tokens)?
>
> This is an interesting point. We built off from probes, e.g. Zhang et al. 2025, which operate on the last token’s representation only. We also thought that the attention layer would be expressive enough to aggregate the relevant information from all the other tokens. However, it’s entirely possible that taking a mean across tokens would yield improved performance. If time allows, we will investigate this idea too.
>
> > Does the MLP simply map from \mathbb R to \mathbb R^d?
>
> Yes, that is indeed the case.
>
> > maybe I'm missing it, but it seems like the correctness probability p (section 3.3) is not defined anywhere. Is this simply the predicted \tilde y?
>
> Thank you for catching this mistake. We should write $\tilde y$ instead of $p$.
>
> > the training set seems to be very small (224 problems, cf. line 730). Could the authors elaborate on why the training set is chosen so small? How does the validation error look like during training, how long does training take, etc.? If MSV can be trained quickly, this could be highlighted as an advantage in the paper.
>
> __Table R.17__ Brier score of Probe on AIME, depending on the number of problems in the training dataset.
>
> | # of problems | Brier score |
> |-------------:|------------:|
> | 56           | 0.0833      |
> | 112          | 0.0821      |
> | 224          | 0.0809      |
> | 448          | 0.0812      |
>
> In our preliminary experiments with Probe in the Terminal Answers setting, we found that the performance stays about the same when doubling the number of training problems from 224 to 448. We chose to use 224 problems for this reason, and also for the reason that it allows us to perform experiments rather quickly. We would also like to note that, since we sample 64 model responses per problem, this creates 224 x 64 = 14,336 sequences in our training set, which is much larger than the 1,000 sequences used by Zhang et al., 2025 in training their probes.
>
> https://anonymous.4open.science/api/repo/msv-rebuttal-iclr2026-016E/file/cGUP/bon.jpg \
> https://anonymous.4open.science/api/repo/msv-rebuttal-iclr2026-016E/file/cGUP/nll.jpg
>
> The above are plots of validation NLL and validation best-of-64 during training of MSV 64 for 4 epochs. We can see that best-of-64 saturates at around the first epoch.
>
> > line 81: "... that latency doesn't grow" -> This claim doesn't seem to be supported anywhere (in particular, I was not able to find latency tradeoffs as N grows). Also, if latency doesn't grow, scaling beyond $N=64$ would be interesting to see.
>
> __Table R.18__ End-to-end latency of early stopping required to achieve fixed target accuracy, assuming that the number of GPUs scales proportionally with the number of sequences $N$. Measured on the MATH dataset with Probe as the verifier.
> | N  | Latency (s) |
> |---:|------------:|
> | 1  | 51.8        |
> | 4  | 25.8        |
> | 16 | 22.9        |
> | 64 | 17.6        |
> | 128| 19.6        |
>
> Thank you for pointing this out. This was a claim made for the ideal situation where we can scale the number of GPUs proportionally to the number of sequences $N$. Table R.18 reports the end-to-end latency in that situation, when using Probe on the MATH dataset, and the threshold $\lambda$ is set so that the accuracy matches or surpasses pass@1 accuracy. We see that the latency required to achieve this fixed accuracy shrinks with increasing $N$. However, at $N=64$, the latency saturates, which is not surprising given that best-of-N performance also saturates at large $N$. [3]
>
> Thank you again for your valuable insights. We've incorporated revisions to our manuscript based on your feedback, and would be happy to discuss any further questions that you may have.
>
> [1] Błasiok et al., 2023. “When Does Optimizing a Proper Loss Yield Calibration?” https://arxiv.org/abs/2305.18764. \
> [2] Zhang et al., 2025. “The Lessons of Developing Process Reward Models in Mathematical Reasoning” \
> [3] Liu et al., 2025. “Can 1B LLM Surpass 405B LLM? Rethinking Compute-Optimal Test-Time Scaling”

---

> ### Author Response · Authors · 2025-11-26
>
> As mentioned in our response to one of the questions, we performed some additional experiments with a variant of MSV that takes a mean of hidden states across answer tokens instead of using only the last answer token, for obtaining the logit. In the Terminal Answers setting, we compared the performance of MSV$_N$ against its variant, on the AIME dataset. The results are as follows:
>
> __Table R.19__ Brier Score of MSV$_N$ with Last Token vs. Mean on AIME
>
> | Method          | N=1   | N=4  | N=16  | N=64 |
> |-----------------|-------:|-------:|-------:|-------:|
> | MSV N | 0.0819 | 0.0687 | 0.0572 | 0.0387 |
> | MSV N (Variant) | 0.0756 | 0.0641 | 0.0544 | 0.0382 |
>
> We can see that the mean variant provides a slight improvement in performance! It should be noted, however, that the variant also uses slightly more computation, as it requires querying the attention with all the answer tokens, and not just the last answer tokens. We have included this result in the appendix of the revised paper.

---

> ### Author Response · Authors · 2025-11-26
>
> We thank you again for the insightful questions and feedbacks. We have posted a response to your concerns and a revised version of our manuscript. We would greatly appreciate any further feedback or remaining concerns you might have.

---

### Official Review · Reviewer_5z5v · 2025-10-30

**Soundness:** 3
**Presentation:** 3
**Contribution:** 2
**Rating:** 4
**Confidence:** 4

**Summary:**

This paper aims to address two key bottlenecks in parallel test-time scaling for large language models (LLMs)—accurate selection among multiple candidate solutions and high inference latency—by introducing the Multi-Sequence Verifier (MSV). This improved calibration directly enhances best-of-N selection accuracy and enables more reliable confidence estimation. The authors further propose a streaming variant of MSV that supports a novel parallel early-stopping framework: by evaluating intermediate outputs from all sequences simultaneously during parallel decoding, decoding can terminate as soon as any sequence reaches a confidence threshold.

**Strengths:**

Overall, the work makes three key contributions:
 (1) a novel design of MSV as the multi-sequence joint verifier,
(2) the demonstration that superior verifier calibration directly improves parallel scaling performance,
 (3) the introduction of a practical, low-latency parallel early-stopping framework enabled by streaming MSV, which fundamentally rethinks how test-time compute can be scaled without proportional latency costs.

**Weaknesses:**

Overall, the paper presents a moderately novel approach but falls short of a truly significant conceptual leap. The work proposes two main contributions: (i) an improved verifier for best-of-N selection, and (ii) a streaming early-stopping framework.

(1) For the first, the core innovation lies in explicitly incorporating the proportion of sequences that produce symbolically equivalent answers (i.e., consensus frequency) as an auxiliary feature, while still relying on standard attention mechanisms to model cross-sequence interactions. While this integration of frequency-based signals is sensible, it builds incrementally on existing ideas like self-consistency rather than introducing a fundamentally new paradigm.

(2)Regarding the second contribution—the streaming answer setting—the motivation is strong and practically relevant. However, the implementation is limited: intermediate answers are extracted only when a predefined delimiter token (e.g., “Wait”) appears, which relies on ad hoc prompting rather than genuine analysis of the model’s internal reasoning states or semantic completeness. A more principled approach would involve detecting answer readiness from the model’s latent representations or logical progression, not just surface-level trigger tokens.

(3) The experimental evaluation is somewhat narrow. The authors evaluate only a single base LLM (DeepSeek-R1-Distill-Qwen-1.5B), which limits the generalizability of the findings. More importantly, the baselines omit Self-Consistency—a standard and highly relevant method in this domain that selects answers based purely on occurrence frequency without any verifier. This omission makes it difficult to assess whether the gains from MSV truly stem from its joint modeling capability or simply from using any verifier at all. Additionally, given the rapid progress in open-source LLMs, it would strengthen the paper to include stronger and more recent models (e.g., Qwen3 or Llama-3.1) as base generators to demonstrate the robustness of MSV across architectures.

(4) The comparison is not entirely fair with respect to the paper’s central claim of addressing high inference latency. While the paper reports accuracy–latency trade-off curves (e.g., Figure 7), it lacks concrete wall-clock timing measurements (e.g., milliseconds per query) that account for the full pipeline—including MSV’s own inference overhead. Since MSV processes all N sequences jointly through a multi-mask Transformer, its computational cost could be non-negligible, especially as N grows. The paper should explicitly report: (a) the end-to-end latency of MSVₙ vs. baselines (including verifier runtime), and (b) how much additional latency MSV incurs to achieve its accuracy gains. Without this, the efficiency claims remain partially unsubstantiated.

**Questions:**

1.Why is the equivalence relation (∼) assumed to be perfect?
The method relies heavily on symbolic equivalence (e.g., via SymPy) to define answer identity and compute γ. But in many real-world tasks (e.g., open-ended QA, code generation, or non-math reasoning), such a deterministic equivalence checker may not exist. How would MSV generalize to domains where answer equivalence is fuzzy or subjective?

2.How sensitive is MSV to the choice of delimiter token (“Wait”) in the Streaming setting?

---

> ### Author Response · Authors · 2025-11-19
>
> We appreciate your thoughtful questions and feedback.
>
> ### W1. Relation to self-consistency
>
> > the baselines omit Self-Consistency
>
> We apologize that this was not highlighted clearly in the main text. **Appendix C** contains detailed experiments with self-consistency (See Table 15), which show that it heavily underperforms compared to best-of-N across all benchmarks. As an example, Table R.14 shows a comparison on the AMC 12 dataset.
>
> __Table R.14__ Comparison between best-of-N and self-consistency on AMC 12
>
> | Method                         | Pass@1 |
> |--------------------------------|-------:|
> | Pass@1 (average accuracy)      | 0.3965 |
> | Self-consistency@64            | 0.4403 |
> | Best-of-64 with MSV 64         | 0.5866 |
>
>
> > it builds incrementally on existing ideas like self-consistency rather than introducing a fundamentally new paradigm
>
> Self-consistency has a critical limitation in that it relies solely on answer frequency, and cannot make use of powerful verifiers. This limitation is empirically manifest in our experiments and the literature (Snell et al., 2024), where self-consistency always performs worse than best-of-N. Weighted voting (WV; Li et al., 2022) aims to combine self-consistency with the power of trained verifiers, by giving more weight to answers with both high answer frequency *and* high verifier score. However, to combine the two, WV uses a rule-based heuristic, rendering its benefits in performance highly dependent on the dataset, as evidenced in our main experiments.
>
> Our method, Multi-Sequence Verifier (MSV), stands out as an architecture that *learns* to make use of global features such as answer frequency, in an end-to-end manner. In particular, MSV attends to all sequences with a transformer architecture, allowing it to use cross-sequence information that's not limited to answer frequency. Empirically, we have extensively shown in our paper that MSV provides a significant advantage over self-consistency and weighted-voting.
>
> ### W2. Ad-hoc-ness of a predefined delimiter token
>
> > intermediate answers are extracted only when a predefined delimiter token (e.g., “Wait”) appears, which relies on ad hoc prompting
>
> > A more principled approach would involve detecting answer readiness from the model’s latent representations
>
> We appreciate this interesting idea. We followed Zhang et al., 2025 and Yang et al., 2025 in using delimiters to separate reasoning blocks, because it was sufficient for our purpose of demonstrating early stopping. Training a separate head to detect the start of a new reasoning block would indeed be an interesting alternative that could provide an even better accuracy latency tradeoff.
>
> > How sensitive is MSV to the choice of delimiter token (“Wait”) in the Streaming setting?
>
> __Table R.15__ Area Under the Tradeoff Curve with Different Delimiters, on the AIME dataset, N=16 parallel sequences
>
> | Delimiter     | Probe   | MSV 1  | MSV 16  |
> |---------------|--------:|-------:|--------:|
> | Wait          | 1024.2  | 954.8  | **1078.3**  |
> | Alternatively  | 985.9   | 980.0  | **1012.4**  |
> | But           | 1039.3  | 1023.6 | **1070.7**  |
>
>
> We provide ablations with two other delimiters, “Alternatively” and “But”, which appear frequently as the first token of a paragraph. Table R.15 reports the area under the tradeoff curve (AUTC) with different delimiters and verifiers. The results show that the delimiter does have some effect on the tradeoff, but the ordering between the three verifiers stays the same, where MSV 16 always remains superior.
>
>
> ### W3. Limited experiments with one base LM
>
> > The authors evaluate only a single base LLM (DeepSeek-R1-Distill-Qwen-1.5B)
> >  it would strengthen the paper to include stronger and more recent models (e.g., Qwen3 or Llama-3.1) as base generators
>
> We appreciate your concern. We provide experiments with different base LMs in **Common Response 1**.
>
> ### W4. Lack of wall-clock timing measurements
>
> >  it lacks concrete wall-clock timing measurements (e.g., milliseconds per query) that account for the full pipeline—including MSV’s own inference overhead.
>
> We appreciate your concern. We provide a detailed report of latency in **Common Response 2**.

---

> ### Author Response · Authors · 2025-11-19
>
> ### Responses to Questions
>
> > The method relies heavily on symbolic equivalence (e.g., via SymPy) to define answer identity and compute γ. But in many real-world tasks (e.g., open-ended QA, code generation, or non-math reasoning), such a deterministic equivalence checker may not exist.
>
> Thank you for raising this important point. As is common in the existing works using verifiers (Cobbe et al., 2021; Li et al., 2022; Wang et al. 2022; Zhang et al., 2024) we solely focus on verifiable problems with equivalence relations. Extending MSV to fuzzy problems is an interesting direction for future work. One possibility would be to augment the model with alternative features such as embedding similarities, and let MSV learn how to weight these signals end-to-end. We leave a systematic study of such fuzzy-equivalence extensions to future work.
>
> > Why is the equivalence relation (∼) assumed to be perfect?
>
> MSV does not rely on the equivalence relation being perfect; it only needs a checker that provides a reasonably informative (but possibly noisy) signal about when answers should be treated as identical. In fact, this is one conceptual advantage over hard-coded heuristics such as self-consistency or weighted voting: MSV can learn how to best use an imperfect checker during training, rather than assuming its outputs are error-free.
>
> Thank you again for your valuable insights. We've incorporated revisions to our manuscript based on your feedback, and would be happy to discuss any further questions that you may have.

---

> ### Author Response · Authors · 2025-11-26
>
> We thank you again for the insightful questions and feedbacks. We have posted a response to your concerns and a revised version of our manuscript. We would greatly appreciate any further feedback or remaining concerns you might have.

---

### Official Review · Reviewer_SzPu · 2025-11-01

**Soundness:** 3
**Presentation:** 3
**Contribution:** 3
**Rating:** 6
**Confidence:** 4

**Summary:**

The paper argues that the existing verification-based methods either suffer from poor accuracy or use too much inference compute because they wait for the full solutions from the generator. To fix these issues, the paper proposes a multi-sequence verifier, a technique that processes all the candidate solutions together and terminates early even if the solution is not fully generated. In particular, there are two variants of the method: terminal answers and streaming method. Further, the framework applies a multi-mask strategy to capture interactions between multiple solutions, final answers, and equivalent answers.

**Strengths:**

1. The paper tackles an important problem of lack of accurate, calibrated, and fast inference with existing verifiers. To mitigate this, the paper proposes multi-mask training in the final answer and streaming answer scenarios.

2. Ultimately, the proposed method provides decent performance improvements across diverse evaluation benchmarks. Further, it seems to be working better than pertinent baselines such as MSV_1, and Probe.

3. The paper also shows that the MSV achieves better calibration and accuracy-latency tradeoff.

**Weaknesses:**

1. The experiments are performed with just one model size and model family i.e., deepseek-r1-distill-qwen-1.5B.  It would be better to try the method on more models and at various sizes.

2. It feels that having many attention masks that operate on similar sequences is a bit of an overkill. If you have enabled full attention (every sequence attends to every other thing), it remains unclear why other attention masks are needed in practice. There is no ablation which shows that each attention mask adds something to the performance.

3. While it is fascinating to attend to many solutions at a time, I think there are scalability issues with this paradigm. Existing thinking models can generate upto 16K tokens and you can’t control whether the first final answer occurs. The context length and latency will blow up pretty quickly in such scenarios if the number of solutions is in the order of 100s. Whereas, the vanilla verification can operate on solutions independently and start performing well.

**Questions:**

Mentioned above

---

> ### Author Response · Authors · 2025-11-19
>
> We appreciate your thoughtful questions and feedback.
>
> ### W1. Lack of experiments with other base language models
>
> > The experiments are performed with just one model size and model family i.e., deepseek-r1-distill-qwen-1.5B.
>
> Thank you for raising this point. We performed experiments with other base language models, and report them in **Common Response 1**.
>
> ### W2. Ablation of masks in MSV
>
> > There is no ablation that shows that each attention mask adds something to the performance. It remains unclear why other attention masks are needed in practice.
>
> Thank you for raising this important point. We performed ablation experiments on the masks, which is reported in **Common Response 3**.
>
> ### W3. Scalability of MSV
>
> > I think there are scalability issues with this paradigm … The context length and latency will blow up pretty quickly in such scenarios if the number of solutions is in the order of 100s.
>
> We very much appreciate this insightful question. In the Streaming Answers setting, there are $N$ sequences being decoded in parallel. If we assume that delimiters are uniformly distributed throughout a sequence, then the total number of delimiters in all sequences, i.e. the total number of forward passes through MSV, is $O(NL)$ where $L$ is the average length of a sequence. The computational cost of each forward pass is linear in the number of (answer) tokens it attends to, which is also $O(NL)$ (It is not quadratic because we only need to query the attention with the last token of the most recent answer). This yields a total MSV cost of $O(N^2 L^2)$, whereas decoding of the $N$ sequences itself scales as $O(N L^2)$. Thus, MSV $N$ adds an extra factor in $N$, while the dependence on $L$ matches that of decoding.
>
> One simple but effective remedy is to use MSV $M$ with $N$ parallel sequences, where $M$ is kept constant as we scale $N$. This is done by partitioning the N sequences into $N/M$ groups each of size $M$, and feeding in each group through MSV $M$. This reduces the computational complexity from $O(N^2 L^2)$ to $O(MNL^2)$, or $O(NL^2)$ if we ignore $M$ as a constant.
>
> __Table R.13__ Area Under the Tradeoff Curve on the AIME dataset, N=128 parallel sequences
>
> |   | Probe | MSV 1 | MSV 4 | MSV 16 | MSV 64 |
> |---:|---:|---:|---:|---:|---:|
> | AUTC | 984.3 | 963.8 | 1060.0 | 1130.6 | 1168.9 |
>
> In Table R.13, we use this exact scheme to report the area under the tradeoff curve (AUTC ↑) in the N=128 setting, on AIME. We see that MSV $M$ provides a better tradeoff than Probe’s “vanilla verification,” even when $M \neq N$. Importantly, because $M$ is fixed and small, MSV’s additional overhead remains at most proportional to the cost of decoding the sequences themselves, while still benefiting from joint reasoning within each group. This makes it practical to scale $N$ beyond 64-128 without incurring a superlinear blow-up in cost relative to standard parallel decoding with vanilla verification.
>
> Thank you again for your valuable insights. We've incorporated revisions to our manuscript based on your feedback, and would be happy to discuss any further questions that you may have.

---

> ### Author Response · Authors · 2025-11-26
>
> We thank you again for the insightful questions and feedbacks. We have posted a response to your concerns and a revised version of our manuscript. We would greatly appreciate any further feedback or remaining concerns you might have.

---

### Author Response · Authors · 2025-11-19

### General Response

We thank the reviewers for their time and effort in reviewing our paper. We are glad that they saw many strengths in our paper. In particular, reviewers noted that the paper is clearly written, well-structured, and easy to follow (R-cGUP). They found the problem setting important and well-motivated (R-SzPu, R-5z5v), and regarded our proposed Multi-Sequence Verifier (MSV) as a novel and interesting contribution (R-5z5v, R-cGUP). Reviewers highlighted the conceptual strength of designing MSV as a joint verifier (R-cGUP), and its use in parallel test-time scaling for better accuracy and reduced latency, while also producing well-calibrated confidence scores (R-SzPu, R-5z5v, R-cGUP). They all agreed that our method empirically achieves promising improvements across diverse benchmarks, outperforming pertinent baselines such as MSV 1 and Probe (R-SzPu, R-5z5v, R-cGUP). We address their main concerns in the three common responses below.

---

> ### Author Response · Authors · 2025-11-19
>
> ### Common Response 1. Different Base Language Models
>
> Reviewers SzPu and 5z5v questioned whether the effectiveness of MSV generalizes beyond the DeepSeek-R1-Distill-Qwen-1.5B base model. To address this, we conducted additional experiments using three other base LMs: DeepSeek-R1-Distill-Llama-8B, Qwen3-1.7B in thinking mode, and Llama-3.2-1B-Instruct. All training hyperparameters were kept the same as in our main experiments.
>
> To test the performance of MSV on larger base models, we performed experiments with DeepSeek-R1-Distill-Llama-8B within the AIME Terminal Answers setup. As shown in Table R.1, MSV $N$ shows improved calibration compared to the baselines, and the improvement scales with $N$. Also, Table R.2 shows that MSV 64 provides a substantial advantage over baselines when used for best-of-64. These results demonstrate that MSV effectively generalizes to larger models.
>
> __Table R.1__ Brier and AUROC of verifiers with DeepSeek-R1-Distill-Llama-8B, on AIME
>
> |  | Probe | MSV 1 | MSV 16 | MSV 64 |
> |---|---|---|---|---|
> | Brier ↓ | 0.0668 | 0.0770 | 0.0298 | 0.0283 |
> | AUROC ↑ | 0.9668 | 0.9746 | 0.9908 | 0.9923 |
>
> __Table R.2__ Best-of-64 accuracy with DeepSeek-R1-Distill-Llama-8B, on AIME
>
> |  | Probe | Probe + WV | MSV 1 | MSV 1 + WV | MSV 64 |
> |---|:---:|:---:|:---:|:---:|:---:|
> | Best-of-64 | 0.6104 | 0.6337 | 0.6052 | 0.6386 | **0.6625** |
>
> Next, to evaluate whether MSV generalizes to more recent reasoning models, we conducted experiments with Qwen3-1.7B in thinking mode, on the AIME dataset. We observe the same trend in Tables R.3 and R.4 as we did with DeepSeek-R1-Distill-Llama-8B.
>
> __Table R.3__ Brier and AUROC of verifiers with Qwen3-1.7B, on AIME
>
> |  | Probe | MSV 1 | MSV 4 | MSV 16 | MSV 64 |
> |---|---:|---:|---:|---:|---:|
> | Brier ↓ | 0.0849 | 0.0770 | 0.0595 | 0.0440 | 0.0317 |
> | AUROC ↑ | 0.9530 | 0.9570 | 0.9696 | 0.9825 | 0.9878 |
>
> __Table R.4__ Best-of-64 accuracy with Qwen3-1.7B, on AIME
>
> |  | Probe | Probe + WV | MSV 1 | MSV 1 + WV | MSV 64 |
> |---|:---:|:---:|:---:|:---:|:---:|
> | Best-of-64 | 0.5223 | 0.5089 | 0.4910 | 0.5294 | **0.5491** |
>
> Finally, to test whether MSV is also effective with non-reasoning models, we ran experiments with Llama-3.2-1B-Instruct. Since its accuracy on AIME is close to zero, we instead trained and evaluated on the MATH train/test splits. Tables R.5 and R.6 also show the same trends, with MSV being clearly superior to the baselines.
>
> __Table R.5__ Brier and AUROC of verifiers with Llama-3.2-1B-Instruct, on MATH
>
> |  | Probe | MSV 1 | MSV 16 | MSV 64 |
> |---|---:|---:|---:|---:|
> | Brier ↓ | 0.1236 | 0.1192 | 0.0596 | 0.0510 |
> | AUROC ↑ | 0.7856 | 0.7654 | 0.9222 | 0.9513 |
>
> __Table R.6__ Best-of-64 accuracy with Llama-3.2-1B-Instruct, on MATH
>
> |  | Probe | Probe + WV | MSV 1 | MSV 1 + WV | MSV 64 |
> |---|:---:|:---:|:---:|:---:|:---:|
> | Best-of-64 | 0.2393 | 0.3402 | 0.1732 | 0.3402 | **0.3778** |

---

> ### Author Response · Authors · 2025-11-19
>
> ### Common Response 2. Latency Analysis
>
> Reviewers 5z5v and cGUP noted that the paper does not provide a detailed report of the actual latency in the Streaming Answers setting. To substantiate our main claim that MSV improves the accuracy-latency tradeoff compared to baselines, we report detailed measurements of latency on the MATH dataset. To be specific, for each verifier and $N$, we measure the minimum latency at which the respective accuracy (on the verifier’s accuracy-latency curve) matches or surpasses the highest accuracy achievable by Probe, which we call the target accuracy. As requested by reviewers, we break down the latency into three components and report each separately: chain-of-thought generation, intermediate answer generation, and verifier inference.
>
> __Table R.7__ Token counts required to achieve the target accuracy
>
> | N  | Probe  | MSV N  |
> |----:|-------:|-------:|
> | 4   | 3699.5 | 2886.3 |
> | 16  | 4022.8 | 1616.4 |
> | 64  | 4131.8 | 1407.4 |
>
> First, Table R.7 reports the token positions at which each verifier reaches the target accuracy at $N$. As we had already shown in our paper, MSV $N$ achieves the same accuracy as Probe using *much fewer tokens*.
>
> __Table R.8__ Average latency of CoT generation, in seconds
>
> | N  | Probe  | MSV N |
> |----:|:------:|:-----:|
> | 4  | 87.5   | 68.3  |
> | 16 | 175.4  | 70.4  |
> | 64 | 568.7  | 193.7 |
>
> __Table R.9__ Average latency of intermediate answer generation, in seconds
>
> | N  | Probe | MSV N |
> |----:|:-----:|:-----:|
> | 4  | 0.9   | 0.7   |
> | 16 | 1.9   | 0.7   |
> | 64 | 6.2   | 2.1   |
>
> __Table R.10__ Average latency of inference with verifier, in seconds
>
> | N  | Probe   | MSV N |
> |----:|:-------:|:-----:|
> | 4  | 0.0003  | 0.1   |
> | 16 | 0.001   | 0.4   |
> | 64 | 0.005   | 3.1   |
>
> __Table R.11__ Average end-to-end latency, in seconds
>
> | N  | Probe  | MSV N |
> |----:|:-----:|:-----:|
> | 4  | 88.5  | **69.2**  |
> | 16 | 177.3 | **71.6**  |
> | 64 | 575.0 | **199.0** |
>
> Next, we report the actual wall-clock time for each of the three components — chain-of-thought generation, intermediate answer generation, and verifier inference. The corresponding measurements are provided in Tables R.8, R.9, and R.10, all obtained using batched computation within a single A6000 GPU. We then sum up these components to obtain the end-to-end latency, as shown in Table R.11. MSV $N$ clearly achieves lower latency than Probe in practice.

---

> ### Author Response · Authors · 2025-11-19
>
> ### Common Response 3. Ablation of Masks
> __Table R.12__ Mask ablation results on AIME, with MSV 16
>
> | Metric | MSV 16 | w/o full | w/o equivalence | w/o within-sequence | w/o within-answer |
> |---:|---:|---:|---:|---:|---:|
> | Brier ↓ | **0.038** | 0.040 | 0.045 | 0.048 | 0.050 |
> | AUROC ↑ | **0.953** | **0.953** | 0.922 | 0.945 | 0.949 |
> | NLL ↓ | **0.150** | 0.157 | 0.236 | 0.192 | 0.169 |
>
> Reviewers SzPu and cGUP expressed concerns that the four masks used in MSV might be redundant, and requested an ablation study. Table R.12 reports the Brier score, AUROC, and NLL of the Streaming MSV 16 model, with each mask ablated. The results show that all masks help, but the equivalence and within-sequence masks are especially important; removing the full mask has the smallest effect.
>
> **Motivation for Multiple Masks**
>
> Our decision to use multiple masks was informed by a preliminary experiment in which we allowed the verifier to attend to all tokens in the sequence, including the chain-of-thought tokens (unlike in the experiments reported in the paper). We initially hypothesized that, since attending to more tokens strictly increases the available information, specialized masks would be unnecessary.
> Contrary to this expectation, we observed that restricting attention to only the answer tokens led to substantially better generalization. This indicated that transformers can be “distracted” by less critical information, and that explicitly guiding attention toward the most informative parts of the sequence can improve performance. Building on this insight, we tested more specialized masks, e.g. masks that attend only to the same answers, and found that appropriately combining these specialized masks yields the strongest overall performance.

---

### Meta-Review · Area_Chair_tCpk · 2026-01-01

**Summary:**

This paper proposes Multi-Sequence Verifier, utilizing joint cross-sequence attention to score parallel candidates and a streaming "Wait" delimiter for early stopping, aiming to improve calibration and reduce test-time latency.

**Reviewer Concerns:**

The authors successfully provided wall-clock latency breakdowns (Reviewers 5z5v, cGUP) and demonstrated generalization across different base models (Reviewers SzPu, 5z5v). However, fundamental flaws remain unaddressed:

1). Incremental Novelty: The method is effectively an over-engineered Self-Consistency utilizing consensus signals. Reliance on ad-hoc "Wait" delimiters constitutes fragile prompt engineering, validated by sensitivity to token choice (Reviewer 5z5v).

2). Limited Scope: The strict dependence on symbolic equivalence (SymPy) renders the method inapplicable to general-purpose reasoning or fuzzy domains (Reviewer 5z5v).

3). Scalability Bottlenecks: The proposed "grouping" workaround for large $N$ effectively admits the native architecture fails to scale efficiently without complexity blowup (Reviewer SzPu).

4). Overfitting Risks: Training on a micro-dataset of only 224 problems raises significant concerns regarding generalization beyond the specific competition benchmarks (Reviewer cGUP).

**Reviewer Scores:**

Reviewer 5z5v is likely to retain the negative score, as the rebuttal confirmed the "Wait" token is an ad-hoc hack and the method lacks broader applicability.

Reviewer SzPu and Reviewer cGUP are unlikely to champion the paper. The marginal utility of the complex masking scheme and the tiny training set suggest the method seems a niche, over-fitted optimization rather than a robust improvement.

---

### Decision · Program_Chairs · 2026-01-26

Reject